# Differences in the Accessory Genomes and Methylomes of Strains of *Streptococcus equi* subsp. *equi* and of *Streptococcus equi* subsp. *zooepidemicus* Obtained from the Respiratory Tract of Horses from Texas

Ellen Ruth A. Morris,[a] Jing Wu,[b] Angela I. Bordin,[a] Sara D. Lawhon,[b] Noah D. Cohen[a]

[a]Department of Large Animal Clinical Sciences, College of Veterinary Medicine & Biomedical Sciences, Texas A&M University, College Station, Texas, USA
[b]Department of Veterinary Pathobiology, College of Veterinary Medicine & Biomedical Sciences, Texas A&M University, College Station, Texas, USA

**ABSTRACT** *Streptococcus equi* subsp. *equi* (SEE) is a host-restricted equine pathogen considered to have evolved from *Streptococcus equi* subsp. *zooepidemicus* (SEZ). SEZ is promiscuous in host range and is commonly recovered from horses as a commensal. Comparison of a single strain each of SEE and SEZ using whole-genome sequencing, supplemented by PCR of selected genes in additional SEE and SEZ strains, was used to characterize the evolution of SEE. But the known genetic variability of SEZ warrants comparison of the whole genomes of multiple SEE and SEZ strains. To fill this knowledge gap, we utilized whole-genome sequencing to characterize the accessory genome elements (AGEs; i.e., elements present in some SEE strains but absent in SEZ or vice versa) and methylomes of 50 SEE and 50 SEZ isolates from Texas. Consistent with previous findings, AGEs consistently found in all SEE isolates were primarily from mobile genetic elements that might contribute to host restriction or pathogenesis of SEE. Fewer AGEs were identified in SEZ because of the greater genomic variability among these isolates. The global methylation patterns of SEE isolates were more consistent than those of the SEZ isolates. Among homologous genes of SEE and SEZ, differential methylation was identified only in genes of SEE encoding proteins with functions of quorum sensing, exopeptidase activity, and transitional metal ion binding. Our results indicate that effects of genetic mobile elements in SEE and differential methylation of genes shared by SEE and SEZ might contribute to the host specificity of SEE.

**IMPORTANCE** Strangles, caused by the host-specific bacterium *Streptococcus equi* subsp. *equi* (SEE), is the most commonly diagnosed infectious disease of horses worldwide. Its ancestor, *Streptococcus equi* subsp. *zooepidemicus* (SEZ), is frequently isolated from a wide array of hosts, including horses and humans. A comparison of the genomes of a single strain of SEE and SEZ has been reported, but sequencing of further isolates has revealed variability among SEZ strains. Thus, the importance of this study is that it characterizes genomic and methylomic differences of multiple SEE and SEZ isolates from a common geographic region (*viz.*, Texas). Our results affirm many of the previously described differences between the genomes of SEE and SEZ, including the role of mobile genetic elements in contributing to host restriction. We also provide the first characterization of the global methylome of *Streptococcus equi* and evidence that differential methylation might contribute to the host restriction of SEE.

**KEYWORDS** DNA methylation, methylome, *Streptococcus equi*, whole-genome sequencing

Address correspondence to Noah D. Cohen, ncohen@cvm.tamu.edu.
The authors declare no conflict of interest.

Streptococcus equi subspecies *equi* (SEE) is a host-restricted pathogen (1–5) and the causative agent of the infectious disease strangles. An ancient and highly contagious upper respiratory disease of horses, strangles is characterized by swollen lymph

nodes, purulent nasal discharge, guttural pouch empyema, lethargy, and fever (2, 4). SEE is thought to have evolved from an ancient strain of *Streptococcus equi* subspecies *zooepidemicus* (SEZ) through a proposed evolutionary bottleneck (6–8). Generally, SEZ is an opportunistic pathogen of horses (9) and is commonly recovered from the respiratory tract as a commensal bacterium (10); however, strains of SEZ are known to cause outbreaks of upper respiratory tract disease resembling strangles in horses (11, 12). SEZ is also a pathogen of other mammalian species, including livestock and humans (13–17).

Published reports of genomic comparisons of SEE and SEZ are exiguous. Differences between the strains SEE 4047 and SEZ H70 were associated with the acquisition of mobile genetic elements such as integrative conjugative elements (ICE) and prophages (6). Specifically, SEZ H70 was described to have 2 ICEs and no acquired prophage, whereas 2 ICEs (ICE*Se1*, ICE*Se2*) and 4 prophages (φSeq1 to φSeq4) were found in SEE 4047. Using quantitative PCR to compare genes identified in either the SEE 4047 or SEZ H70 genome with additional isolates of SEZ and SEE, that study further indicated that evolution of SEE from SEZ was associated with reduced genetic diversity in SEE. (6). Greater genetic variation has been described for isolates of SEZ than for those of SEE (14, 18). Furthermore, the multilocus sequencing typing (MLST) database has characterized over 400 sequence types (ST) of SEZ, whereas only 2 primary STs have been described for SEE (accessed 6 February 2021) (7). Thus, it is likely that the mobile genetic elements unique to SEE have contributed to gene losses or duplications that have contributed to constraining SEE to a single host, whereas greater genetic diversity of SEZ might explain its ability to adapt to many mammalian hosts. Much remains to be learned, however, about the differences between SEE and SEZ and about how SEE evolved to be host restricted.

Very limited data comparing strains of SEZ from the respiratory tract of horses with clinical isolates of SEE from horses using untargeted sequencing methods such as whole-genome sequencing (WGS) are available to substantiate existing observations (6). One untargeted approach for studying bacterial species is to define and compare the core and accessory genomes of the individual species. This tack has been described either for studying a single bacterial species or to compare several species of streptococcal organisms (19, 20). The core genome elements for subspecies are defined as those found in the genomes of both subspecies, and the accessory genome elements (AGEs) for subspecies are those that are not found among the subspecies core genome elements. Furthermore, it is possible with PacBio WGS to characterize the complete methylome of prokaryotes (21). Traditionally, the presence of methylation of bacterial DNA has been recognized as a means by which bacteria are protected against bacteriophages or other foreign DNA. Methyl groups present on the same sequence motifs protect against enzymatic degradation, whereas the DNA lacking the same methylation is recognized as foreign by bacterial endonucleases and results in cleavage at these unmethylated motifs (22, 23). Methylation, however, can also alter gene expression, alter virulence in some bacteria (24–26), and even result in adaptive evolution (27). Methylated bacterial DNA is most commonly recognized as residues of N6-methyl-adenosine (m6A), N4-methyl-cytosine (m4C), or C5-methyl-cytosine (m5C) (22, 23). Thus, we used the WGS technology of PacBio single molecule, real-time (SMRT) to characterize the core genome and accessory genomes and to compare the methylomes of SEE and SEZ to identify potential differences that might help elucidate how SEE evolved to become a host-specific pathogen.

## RESULTS

Comparisons of the accessory genome of the 50 SEE and 50 SEZ isolates were performed using the Spine, AGEnt, and ClustAGE pipeline (Fig. 1). The AGEs found only in the 50 SEE isolates were associated with primarily 1 of the 2 ICEs or 1 of the 4 acquired prophages described for SEE 4047 (6), and a total of 85 coding sequences (CDS) within the SEE 4047 genome were identified: 4 of the 85 elements were within the region of the ICE*Se1* elements (SEQ_0756 to SEQ_0758, SEQ_0761) and 36 of the 85 elements

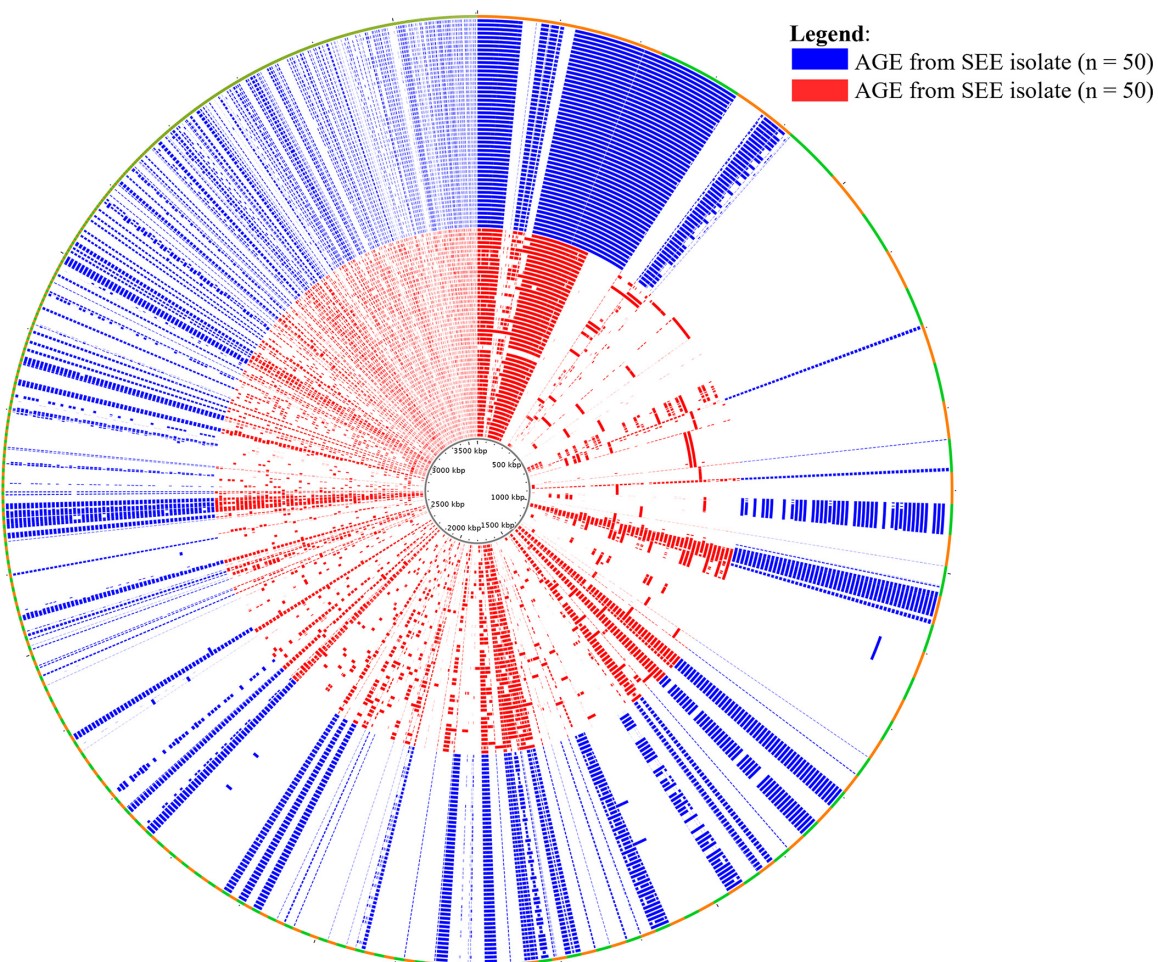

**FIG 1** Comparison of accessory genome elements (AGE) of SEE (*n* = 50) and SEZ (*n* = 50) genomes. The outer ring shows the ClustAGE bins that are ≥200 base pairs in size; these are ordered clockwise from the largest bin to the smallest bin and are differentiated by orange and green to define bin borders. The concentric inner bands show the distribution of AGE within each individual isolate. Bands that are blue represent SEE isolates, and bands that are red represent SEZ isolates. The central ruler of the figure indicates the cumulative size of the AGE in kilobases.

were associated with ICE*Se2* (Table 1). Of the 85 CDS, none (0) of the AGEs were located on prophage φSeq1, 17 were part of φSeq2, 20 were from φSeq3, and 7 were on φSeq4 (Table 1). Finally, SEQ_1102, a site-specific recombinase, was identified as part of the AGEs and was not found on either of the 2 ICEs or 4 prophages but, rather, was associated with an insertion element in SEE 4047. The CDS associated with the ICE and prophages from SEE 4047 were not found in all 50 SEE isolates. The functions of the identified AGEs were associated with primarily those of the acquired prophages and of hypothetical proteins. Additionally, the CDS that comprise the equibactin locus (SEQ_1233 to SEQ_1246) and 3 of the 4 superantigens, *seeH* (SEQ_2036), *seeI* (SEQ_2037), and *seeL* (SEQ_1728), were also identified as part of the AGE of SEE relative to SEZ. Equibactin is a novel locus found on ICE*Se2* in SEE that is involved in iron acquisition (28) and is encoded by the 14 CDS, SEQ_1233 to SEQ_1246. The Gene Ontology (GO) functions and pathway interactions of the 85 CDS identified in the AGEs from SEE were assessed using ClueGO, which characterized 23 CDS (Fig. 2, Table S1). The primary GO functions identified were DNA modification, endonuclease activity, and ATPase activity. Also noted were the KEGG pathways of biosynthesis of the siderophore group nonribosomal peptides and *Staphylococcus aureus* infection.

Next, elements that were specific to all 50 SEZ genomes were considered, and only 15 CDS from the H70 genome were identified (Table 2). Of the 15 CDS, 8 had been previously described to be deleted from the SEE 4047 genome (6), in agreement with our findings.

**TABLE 1** Accessory genome elements identified in all 50 SEE genomes

| RefSeq_4047 | Gene name | Region | PSORTb result | Protein |
|---|---|---|---|---|
| SEQ_0756 | | ICESe1 | Cytoplasmic | Transcriptional regulator |
| SEQ_0757 | | ICESe1 | Cytoplasmic | Modification methylase PstI (EC 2.1.1.72) |
| SEQ_0758 | | ICESe1 | Cytoplasmic | Type II site-specific deoxyribonuclease |
| SEQ_0761 | | ICESe1 | Cytoplasmic | USG protein |
| SEQ_0787 | | Prophage Seq2 | Unknown | Phage integrase: site-specific recombinase |
| SEQ_0816 | | Prophage Seq2 | Unknown | Phage protein |
| SEQ_0817 | | Prophage Seq2 | Unknown | Phage protein |
| SEQ_0818 | | Prophage Seq2 | Unknown | Phage endonuclease |
| SEQ_0819 | | Prophage Seq2 | Cytoplasmic | Phage terminase |
| SEQ_0823 | | Prophage Seq2 | Cytoplasmic | Phage portal protein |
| SEQ_0824 | | Prophage Seq2 | Cytoplasmic | Prophage Clp protease-like protein |
| SEQ_0825 | | Prophage Seq2 | Cytoplasmic | Phage capsid protein |
| SEQ_0826 | | Prophage Seq2 | Cytoplasmic | Putative capsid protein (ACLAME 311) |
| SEQ_0827 | | Prophage Seq2 | Cytoplasmic | DNA packaging protein |
| SEQ_0828 | | Prophage Seq2 | Unknown | Phage protein |
| SEQ_0829 | | Prophage Seq2 | Cytoplasmic | Phage protein |
| SEQ_0830 | | Prophage Seq2 | Cytoplasmic | Phage protein |
| SEQ_0831 | | Prophage Seq2 | Cytoplasmic | Phage major tail protein |
| SEQ_0832 | | Prophage Seq2 | Unknown | Phage protein |
| SEQ_0833 | | Prophage Seq2 | Unknown | Phage protein |
| SEQ_0835 | | Prophage Seq2 | Unknown | Phage-related protein |
| SEQ_1102 | | Insertion element | Cytoplasmic | Site-specific recombinase |
| SEQ_1231 | | ICESe2 | Cytoplasmic | Hypothetical protein |
| SEQ_1233 | eqbN | ICESe2 | Unknown | Hypothetical protein |
| SEQ_1234 | eqbM | ICESe2 | Unknown | Hypothetical protein |
| SEQ_1235 | eqbL | ICESe2 | Cytoplasmic membrane | Heterodimeric efflux ABC transporter |
| SEQ_1236 | eqbK | ICESe2 | Cytoplasmic membrane | Heterodimeric efflux ABC transporter |
| SEQ_1237 | eqbJ | ICESe2 | Cytoplasmic membrane | Duplicated ATPase component BL0693 of energizing module of predicted ECF transporter |
| SEQ_1238 | eqbI | ICESe2 | Cytoplasmic membrane | Transmembrane component BL0694 of energizing module of predicted ECF transporter |
| SEQ_1239 | eqbH | ICESe2 | Cytoplasmic membrane | Substrate-specific component BL0695 of predicted ECF transporter |
| SEQ_1240 | eqbG | ICESe2 | Cytoplasmic | Hypothetical protein |
| SEQ_1241 | eqbF | ICESe2 | Cytoplasmic | Hypothetical protein |
| SEQ_1242 | eqbE | ICESe2 | Cytoplasmic | Polyketide synthase modules and related proteins |
| SEQ_1243 | eqbD | ICESe2 | Cytoplasmic | 2,3-Dihydroxybenzoate-AMP ligase (EC 2.7.7.58) of siderophore biosynthesis |
| SEQ_1244 | eqbC | ICESe2 | Cytoplasmic | 4′-Phosphopantetheinyl transferase (EC 2.7.8.-) |
| SEQ_1245 | eqbB | ICESe2 | Cytoplasmic | Iron acquisition yersiniabactin synthesis enzyme YbtT @ thioesterase in siderophore biosynthesis gene cluster |
| SEQ_1246 | eqbA | ICESe2 | Cytoplasmic | Iron-dependent repressor |
| SEQ_1249 | | ICESe2 | Unknown | Hypothetical protein |
| SEQ_1250 | | ICESe2 | Cytoplasmic | Hypothetical protein |
| SEQ_1252 | | ICESe2 | Cytoplasmic | Hypothetical protein |
| SEQ_1253 | | ICESe2 | Cell wall/extracellular | Superfamily II DNA and RNA helicase |
| SEQ_1254 | | ICESe2 | Cytoplasmic | Hypothetical protein |
| SEQ_1257 | | ICESe2 | Cytoplasmic | FIG00645039: hypothetical protein with HTH-domain |
| SEQ_1258 | | ICESe2 | Cytoplasmic | Abortive infection protein AbiGI |
| SEQ_1260 | | ICESe2 | Unknown | Hypothetical protein |
| SEQ_1261 | | ICESe2 | Unknown | NLP/P60 family protein |
| SEQ_1262 | | ICESe2 | Cytoplasmic | Modification methylase Cfr9I (EC 2.1.1.113) |
| SEQ_1263 | | ICESe2 | Unknown | TrsE-like protein |
| SEQ_1264 | | ICESe2 | Cytoplasmic membrane | Hypothetical protein |
| SEQ_1265 | | ICESe2 | Cytoplasmic | Hypothetical protein |
| SEQ_1266 | | ICESe2 | Cytoplasmic membrane | Hypothetical protein |
| SEQ_1267 | | ICESe2 | Cytoplasmic membrane | Maff2 family protein |

**TABLE 1** (Continued)

| RefSeq_4047 | Gene name | Region | PSORTb result | Protein |
|---|---|---|---|---|
| SEQ_1268 | | ICESe2 | Cytoplasmic membrane | Hypothetical protein |
| SEQ_1269 | | ICESe2 | Cytoplasmic membrane | ABC-type antimicrobial peptide transport system |
| SEQ_1270 | | ICESe2 | Cytoplasmic membrane | Hypothetical protein |
| SEQ_1271 | | ICESe2 | Cytoplasmic membrane | Hypothetical protein |
| SEQ_1274 | | ICESe2 | Cytoplasmic | Chromosome (plasmid) partitioning protein ParB |
| SEQ_1275 | | ICESe2 | Cytoplasmic membrane | Chromosome (plasmid) partitioning protein ParA |
| SEQ_1728 | *seeL* | Prophage Seq3 | Unknown | Streptococcal pyrogenic exotoxin K (SpeK) |
| SEQ_1739 | | Prophage Seq3 | Cytoplasmic membrane | Phage tail length tape-measure protein |
| SEQ_1740 | | Prophage Seq3 | Unknown | Conserved hypothetical protein - phage associated |
| SEQ_1741 | | Prophage Seq3 | Cytoplasmic | Conserved hypothetical protein - phage associated |
| SEQ_1742 | | Prophage Seq3 | Unknown | Phage major tail protein |
| SEQ_1743 | | Prophage Seq3 | Cytoplasmic | Phage major tail protein |
| SEQ_1744 | | Prophage Seq3 | Cytoplasmic membrane | Structural protein |
| SEQ_1745 | | Prophage Seq3 | Unknown | Phage protein |
| SEQ_1746 | | Prophage Seq3 | Unknown | Phage protein |
| SEQ_1747 | | Prophage Seq3 | Cytoplasmic | Phage protein |
| SEQ_1748 | | Prophage Seq3 | Cytoplasmic | Hypothetical phage protein |
| SEQ_1749 | | Prophage Seq3 | Unknown | Phage major capsid protein |
| SEQ_1750 | | Prophage Seq3 | Cytoplasmic | Phage major capsid protein |
| SEQ_1751 | | Prophage Seq3 | Unknown | FIG01114710: hypothetical protein |
| SEQ_1755 | | Prophage Seq3 | Cytoplasmic | Guanosine-3′ |
| SEQ_1756 | | Prophage Seq3 | Unknown | Hypothetical protein |
| SEQ_1757 | | Prophage Seq3 | Cytoplasmic | Phi Mu50B-like protein |
| SEQ_1758 | | Prophage Seq3 | Cytoplasmic | Phage portal protein |
| SEQ_1762 | | Prophage Seq3 | Unknown | Pleiotropic regulator of exopolysaccharide synthesis |
| SEQ_1763 | | Prophage Seq3 | Cytoplasmic | Chromosome segregation ATPase |
| SEQ_2036 | *seeH* | Prophage Seq4 | Extracellular | Streptococcal pyrogenic exotoxin H (SpeH); toximoron (superantigen) @ exotoxin |
| SEQ_2037 | *seeI* | Prophage Seq4 | Extracellular | Exotoxin |
| SEQ_2038 | | Prophage Seq4 | Unknown | Phage lysin |
| SEQ_2040 | | Prophage Seq4 | Cytoplasmic membrane | Phage holin |
| SEQ_2041 | | Prophage Seq4 | Unknown | Phage holin |
| SEQ_2042 | | Prophage Seq4 | Cytoplasmic | Phage protein |
| SEQ_2043 | | Prophage Seq4 | Unknown | Hypothetical protein |

These elements were found throughout the SEZ genomes and were not primarily localized to any ICE, unlike the SEE-specific AGEs. Twelve of the 15 SEZ-specific AGEs were localized to the cytoplasm ($n = 5$) or the cytoplasmic membrane ($n = 7$), and a single hypothetical protein was predicted to have an extracellular location; the locations of the remaining hypothetical proteins ($n = 2$) were unknown. The apparent function of these AGEs largely points to differences in fermentation of the carbohydrate lactose (SZO_15220 to SZO_15250) and sorbitol (SZO_01750) (Table 2). Using ClueGO to evaluate the function of the 15 CDS, we associated 3 CDS (*lacE*, *lacF*, and *lacG*) with galactose metabolism (Fig. 3, Table S2); no function was identified for the other 12 CDS.

Single nucleotide polymorphisms (SNPs) were assessed with ParSnp to determine the variance of the core genomes within subspecies (SEE and SEZ) relative to a reference strain. The function of SNPs that were identified in the core genomes for ≥75% (i.e., 38 or more of the 50 SEE or SEZ isolates) were evaluated further. The mean variance of the 50 SEE genomes relative to the reference SEE 4047 was 0.00010% (range, 0.0005% to 0.0015%) (Fig. 4A), and the mean variance of the 50 SEZ genomes relative to the reference SEZ H70 was 0.01853% (range, 0.00013% to 0.02527%) (Fig. 4B). Fifteen SNPs found in ≥75% of SEE genomes were identified at 13 unique CDS (Table S3). Three of the CDS (SEQ_1467, SEQ_1672, and SEQ_1694) had SNPs resulting in a synonymous mutation. The CDS

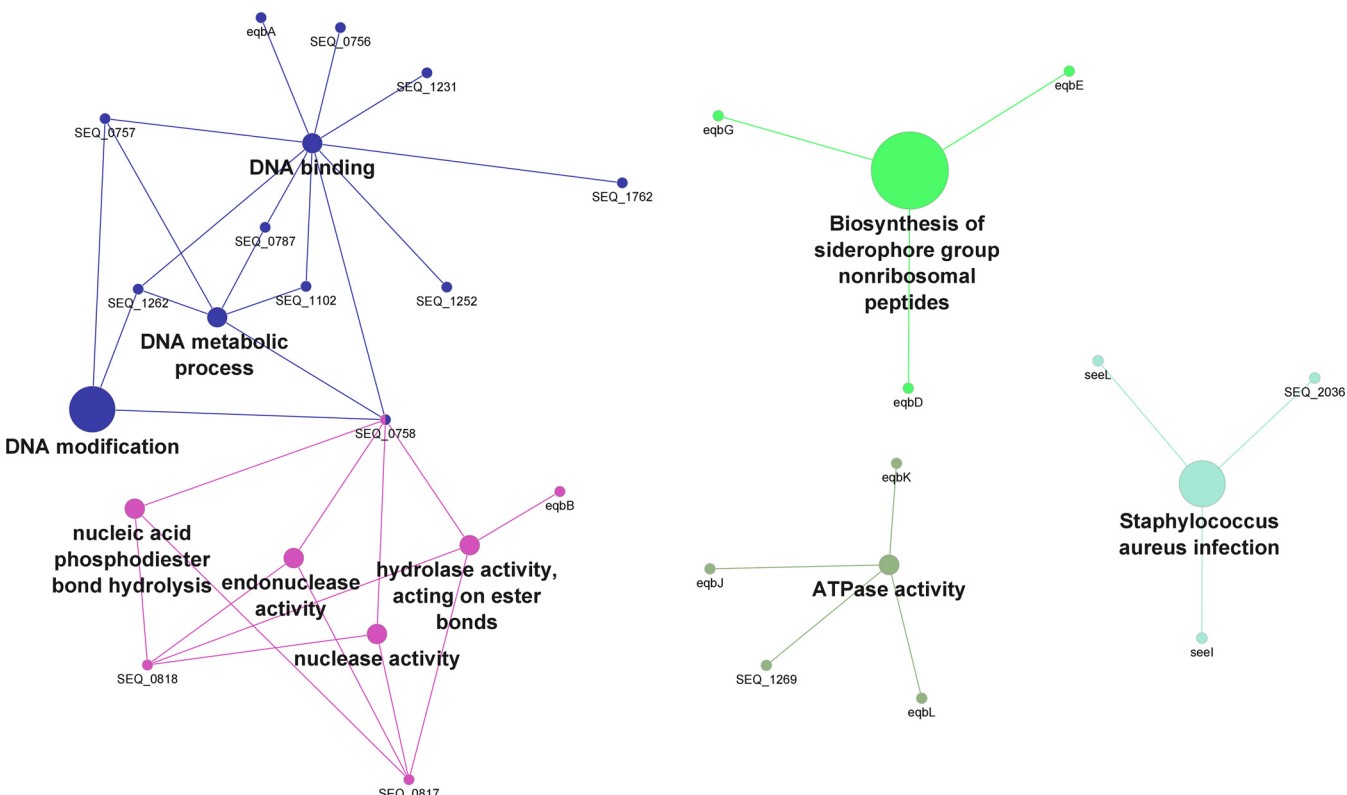

**FIG 2** Gene ontology (GO) terms and KEGG pathways (annotated in ClueGO) in the accessory genome elements identified in all SEE (*n* = 50) genomes. Circle size represents the degree of the positive relationship between the GO terms and the term's adjusted *P* value. The related terms are grouped and presented in the same color.

encoding the SeM protein (SEQ_2017) had 3 different SNPs. Using ClueGO, no GO function or pathway interactions were identified among the functions of the13 SEE CDS in which these 15 SNPs were identified. A total of 587 SNPs found in ≥75% of SEZ genomes were identified at 71 unique CDS (Table S4). The majority of these SEZ SNPs (75%; 441/587) resulted in synonymous mutations. Using ClueGO, we identified interactions of GO functions and pathways for 68 of the 71 SEZ CDS in which SNPs occurred. These function and pathway interactions included transferase activity, nucleic acid binding, cation binding, processing of various molecules, intracellular anatomical structure, membrane and plasma membrane structure, transport of ions, and elements of the 2-component signal transduction pathway (Fig. 5, Table S5).

PacBio SMRT WGS permits characterization of methylation patterns of bacterial genomes through the implementation of the BaseMod pipeline developed by PacBio (21). Using the Restriction Enzyme Database (REBASE) (29), the methylation motifs of a representative subset of 24 isolates each of SEE and SEZ were compared to the reference genomes SEE 4047 and SEZ H70. The methylation motifs identified in the 24 SEE genomes were more consistent than those identified in the 24 SEZ genomes (Table S6). All 24 SEE genomes had the motif sequence CTGCAG with methylation occurring at approximately 95% of each sequencing occurrence. An additional methylation motif, CATCC, was not identified in REBASE but was noted in 13 of 24 SEE isolates, and a single novel methylation motif (GGATGNND) was found in the SEE isolate 18-074 originating from Salado, Texas (Table 3). However, the partnered methylation motif sequences GGATG and CATCC, described in REBASE, were found in only 12 of 24 SEZ isolates. Furthermore, most of the methylation motif sequences recognized in the 24 SEZ isolates were not commonly identified in all isolates, and the majority were novel motifs (Table 3).

Genes encoding homologous proteins from the reference genomes of SEE (4047) and SEZ (H70) with a similarity of ≥99% were selected to compare the presence or

**TABLE 2** Accessory genome elements identified in all 50 SEZ genomes

| RefSeq_H70 | Gene name | Region | PSORTb result | Protein |
|---|---|---|---|---|
| SZO_01750 | *sorD* | Deleted in 4047 | Cytoplasm | Sorbitol-6-phosphate 2-dehydrogenase (EC 1.1.1.140) |
| SZO_14750 | | | Cytoplasm | Transcriptional regulator |
| SZO_15220 | *lacG* | Deleted in 4047 | Cytoplasm | 6-Phospho-beta-galactosidase (EC 3.2.1.85) |
| SZO_15240 | *lacF* | Deleted in 4047 | Cytoplasm | PTS system |
| SZO_15250 | *lacT* | Deleted in 4047 | Cytoplasm | Beta-glucoside bgl operon antiterminator |
| SZO_05610 | | Deleted in 4047 | Cytoplasm membrane | ABC transporter ATP-binding protein |
| SZO_05620 | | Deleted in 4047 | Cytoplasm membrane | Daunorubicin resistance transmembrane protein |
| SZO_05630 | | Deleted in 4047 | Cytoplasm membrane | Efflux ABC transporter |
| SZO_14690 | | ESAT-6-like | Cytoplasm membrane | Branched-chain amino acid transport system carrier protein |
| SZO_14730 | *comB* | | Cytoplasm membrane | Competence-stimulating peptide ABC transporter permease protein ComB |
| SZO_14744 | | | Cytoplasm membrane | Competence-stimulating peptide ABC transporter ATP-binding protein ComA |
| SZO_15230 | *lacE* | Deleted in 4047 | Cytoplasm membrane | PTS system |
| SZO_14742 | | | Extracellular | FIG01116836: hypothetical protein |
| SZO_10380 | | | Unknown | FIG01117834: hypothetical protein |
| SZO_14743 | | | Unknown | FIG01120711: hypothetical protein |

absence of methylation between the *Streptococcus equi* subspecies. In considering sites where methylation occurred in the 24 SEE genomes but not in the 24 SEZ genomes on homologous proteins, 37 CDS were identified. These 37 CDS were identified from a pool of 89 CDS with methylation present in the SEE genomes, and 231 CDS were identified in which SEZ had no methylation. The presence of methylation was found on the motif sequence CTGCAG at 70 different locations within the 37 CDS and was identified as the methylation type m6A (Table 4). To evaluate the GO terms and functions of these 37 CDS, ClueGo was implemented using default parameters. The functions of exopeptidase activity, transition metal ion binding, transmembrane transport, quorum sensing, and propanoate metabolism were identified (Fig. 6, Table S7). Homologous proteins sites where methylation was found in all 24 SEZ genomes but was absent in the 24 SEE genomes were reviewed. Likely due to the variability of SEZ genomes, only 10 potential CDS were identified on homologous proteins (Table S8), and the location and type (m6A or m4A) of methylation were inconsistent among all 24 SEZ genomes.

To determine the influence on gene expression of methylation at the differentially methylated CDS (Table S9), a CDS associated with exopeptidase activity (SEQ_1597) and a CDS associated with quorum sensing (SEQ_1918) were selected to quantify transcription relative to a housekeeping gene (SEQ_1170) (6, 8) using quantitative PCR (qPCR) in a subset of SEE (*n* = 5) and SEZ (*n* = 5) isolates. Reverse transcription of SEQ_1597 revealed a statistically significant (*P* < 0.05) decrease in SEE relative to SEZ

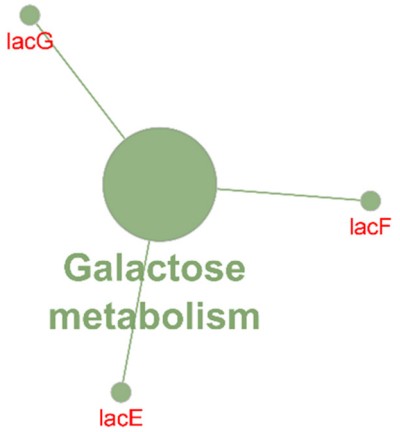

**FIG 3** Gene ontology (GO) terms and KEGG pathways (annotated in ClueGO) in the accessory genome elements identified in all SEZ (*n* = 50) genomes. Circle size represents the degree of the positive relationship between the GO terms and the term's adjusted *P* value. The related terms are grouped and presented in the same color.

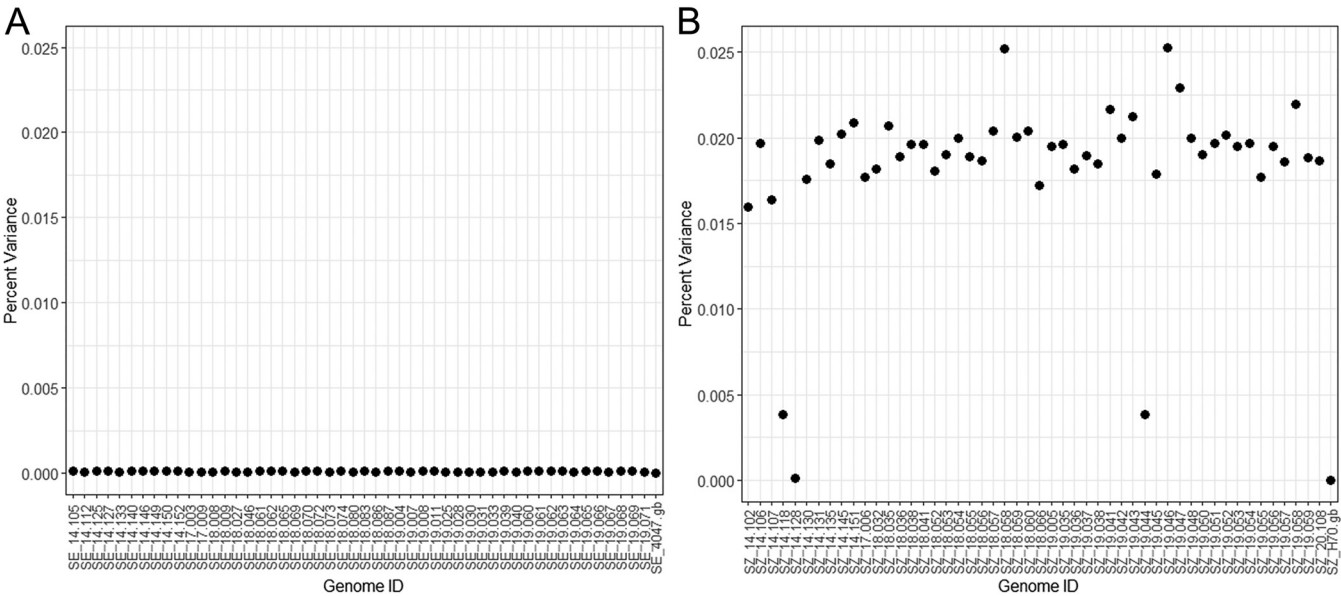

**FIG 4** Percentage of variance for the core genome relative to the reference. (A) Percentage of variance in 50 SEE isolates from Texas. The variance the SEE isolates had a mean of 0.00010% and exhibited a variance range of 0.00005% to 0.00015%. (B) Percentage of variance in 50 SEZ isolates from Texas. The variance of the SEZ isolates had a mean of 0.01853% and demonstrated a range of 0.00013% to 0.02527%.

in the level of transcription relative to the housekeeping gene (SEQ_1170) (Fig. 7). Similarly, reverse transcription of SEQ_1918 demonstrated a level of transcription relative to the housekeeping gene (SEQ_1170) in SEE statistically significantly ($P < 0.05$) higher than that in SEZ (Fig. 7).

## DISCUSSION

Comparison of AGEs among isolates has been used to understand differences within a bacterial species or across genera (19, 20). Our study was designed to help understand which genomic attributes contribute to host specificity of SEE by comparing the AGEs of SEE ($n = 50$) and SEZ ($n = 50$) collected from the respiratory tract of horses from Texas. Analysis of the AGEs analysis identified more SEE-specific CDS than SEZ-specific CDS,

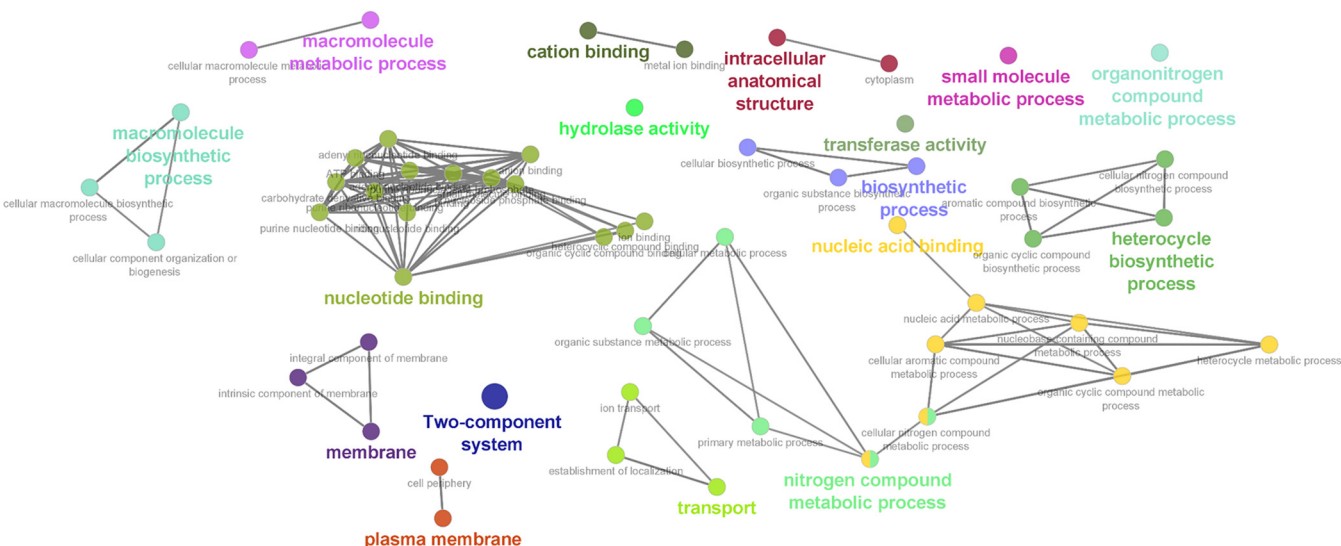

**FIG 5** Gene ontology (GO) terms and KEGG pathways (annotated in ClueGO) for SNPs identified in 75% (≥38) of SEZ genomes. Circle size represents the degree of the positive relationship between the GO terms and the term's adjusted *P* value. The related terms are grouped and presented in the same color.

**TABLE 3** Novel motif sequences from SEE and SEZ genomes

| Motif sequence | Genome ID | Subsp.[a] | Center position | Modification type |
|---|---|---|---|---|
| GGATGNND | 18-074 | *equi* | 3 | m6A |
| ACCNNNNNTCTT/AAGANNNNNGGT | 19-050 | zoo | 4 | m6A |
| ACAYNNNNNRGG | 14-006 | zoo | 3 | m6A |
| ACCCA | 19-052 | zoo | 5 | m6A |
| AGTNNNNNNNGTC/GACNNNNNNNACT | 19-044 | zoo | 1 | m6A |
| AGTNNNNNNNGTC/GACNNNNNNNACT | 19-050 | zoo | 1 | m6A |
| CCANNNNNNNNNTAC/GTANNNNNNNNNNTGG | 18-066 | zoo | 3 | m6A |
| TCANNNNNNNTGG/CCANNNNNNNTGA | 14-151 | zoo | 3 | m6A |
| TCANNNNNNNTGG/CCANNNNNNNTGA | 19-048 | zoo | 3 | m6A |
| CTCCAG/CTGGAG | 18-059 | zoo | 5 | m6A |
| CTCCAG/CTGGAG | 19-043 | zoo | 5 | m6A |
| CTCCAG/CTGGAG | 19-044 | zoo | 5 | m6A |
| CTCCAG/CTGGAG | 18-059 | zoo | 5 | m6A |
| CTCCAG/CTGGAG | 19-043 | zoo | 5 | m6A |
| CTCCAG/CTGGAG | 19-044 | zoo | 5 | m6A |
| GACNNNNNNTARG/CYTANNNNNGTC | 19-047 | zoo | 4 | m6A |
| GACNNNNNNTARG | 19-041 | zoo | 2 | m6A |
| GCANNNNNNNNNTTC/GAANNNNNNNNNNTGC | 19-038 | zoo | 3 | m6A |
| GACNNNNNNTARG | 19-047 | zoo | 2 | m6A |
| GATC | 19-058 | zoo | 2 | m6A |
| GATGC/GCATC | 19-056 | zoo | 2 | m6A |
| GCTANAC | 19-045 | zoo | 6 | m6A |
| TCANNNNNNGTTY/RAACNNNNNNTGA | 18-058 | zoo | 3 | m6A |
| RGATCY | 14-007 | zoo | 5 | m4C |
| RGATCY | 18-055 | zoo | 5 | m4C |
| TCCAG | 17-006 | zoo | 4 | m6A |
| TCCAG | 19-036 | zoo | 4 | m6A |
| YACNNNNNGTR | 19-058 | zoo | 2 | m6A |

[a]zoo, *zooepidemicus*.

demonstrating greater homozygosity (i.e., reduced genetic diversity) in SEE isolates using an untargeted approach. This observation has been described before using the targeted approach of quantitative PCR in isolates from the United Kingdom (6). The AGEs of the SEE isolates were primarily noted to be a part of the prophages ($\varphi$Seq2 to $\varphi$Seq4) and the 2 ICEs (ICE*Se1*, ICE*Se2*) described for the SEE 4047 genome (6). However, no elements of the prophage $\varphi$Seq1 were consistently found in the 50 SEE isolates used in our study. This finding is consistent with comparison of the accessory genome of SEE isolates by Harris et al. (8). It is plausible and probable that these mobile genetic elements mediate gene losses that contribute to the reduced diversity of SEE relative to that of SEZ and that the gene duplications in mobile genetic elements contribute to constraining SEE to a single host. Alternatively, these mobile genetic elements might mediate specific functions that influence host specificity. The elements found on the prophage $\varphi$Seq2 were primarily proteins characterized as phage elements and located in the cytoplasm. The superantigens *seeL* (SEQ_1728), *seeH* (SEQ_2036), and *seeI* (SEQ_2027) located on prophages $\varphi$Seq3 and $\varphi$Seq4 were found in all SEE isolates. In contrast, *seeM* was not identified among our AGEs, which is consistent with evidence of its absence in some strains of SEE (8); *seeM* also has been identified in a small number of strains of SEZ (6). These superantigens have been show *in vitro* to induce increased production of gamma interferon (IFN-$\gamma$) from CD5[+] CD4[+] T-lymphocytes (30). Similarly, superantigens in *Streptococcus pyogenes* (*S. pyogenes*) are described to suppress antibody production, in part due to the production of IFN-$\gamma$ by over-activated CD4[+] T cells (31, 32). The conserved elements on ICE*Se1* ($n = 4$) were proteins denoted as a transcriptional regulator, a modification methylase, a type II site-specific DNase, and a USG protein. These first 3 proteins are identified in REBASE as part of a type II restriction modification system (29), and the function of the USG protein is unknown but it is a member of the SIR protein family (33). The elements from ICE*Se2* ($n = 36$) were hypothetical proteins, transport proteins, the equibactin locus (*eqbA* to *eqbN*), and chromosome partitioning proteins. ICE*Se2* was the most conserved (36 of 85 CDS) of the SEE mobile

**TABLE 4** Methylation location, type, and motif in 24 SEE genomes

| CDS | Location | Type | Motif sequence |
|-----|----------|------|----------------|
| SEQ_0045 | 56855 | m6A | CTGCAG |
| SEQ_0067 | 74697 | m6A | CTGCAG |
| SEQ_0067 | 74700 | m6A | CTGCAG |
| SEQ_0070 | 76364 | m6A | CTGCAG |
| SEQ_0251 | 230695 | m6A | CTGCAG |
| SEQ_0300 | 285354 | m6A | CTGCAG |
| SEQ_0300 | 285357 | m6A | CTGCAG |
| SEQ_0302 | 288740 | m6A | CTGCAG |
| SEQ_0340 | 323013 | m6A | CTGCAG |
| SEQ_0340 | 323016 | m6A | CTGCAG |
| SEQ_0435 | 417164 | m6A | CTGCAG |
| SEQ_0435 | 417167 | m6A | CTGCAG |
| SEQ_0474 | 460537 | m6A | CTGCAG |
| SEQ_0474 | 461395 | m6A | CTGCAG |
| SEQ_0497 | 482852 | m6A | CTGCAG |
| SEQ_0497 | 482855 | m6A | CTGCAG |
| SEQ_0596 | 580039 | m6A | CTGCAG |
| SEQ_0596 | 580042 | m6A | CTGCAG |
| SEQ_0721 | 712040 | m6A | CTGCAG |
| SEQ_0769 | 763220 | m6A | CTGCAG |
| SEQ_0769 | 763223 | m6A | CTGCAG |
| SEQ_0898 | 873274 | m6A | CTGCAG |
| SEQ_0898 | 873277 | m6A | CTGCAG |
| SEQ_0976 | 967141 | m6A | CTGCAG |
| SEQ_1129 | 1118411 | m6A | CTGCAG |
| SEQ_1277 | 1274166 | m6A | CTGCAG |
| SEQ_1277 | 1274169 | m6A | CTGCAG |
| SEQ_1278 | 1276130 | m6A | CTGCAG |
| SEQ_1299 | 1296130 | m6A | CTGCAG |
| SEQ_1299 | 1296133 | m6A | CTGCAG |
| SEQ_1318 | 1318299 | m6A | CTGCAG |
| SEQ_1318 | 1318302 | m6A | CTGCAG |
| SEQ_1407 | 1406622 | m6A | CTGCAG |
| SEQ_1407 | 1406625 | m6A | CTGCAG |
| SEQ_1407 | 1408183 | m6A | CTGCAG |
| SEQ_1410 | 1411644 | m6A | CTGCAG |
| SEQ_1410 | 1411647 | m6A | CTGCAG |
| SEQ_1439 | 1442999 | m6A | CTGCAG |
| SEQ_1439 | 1443002 | m6A | CTGCAG |
| SEQ_1448 | 1453017 | m6A | CTGCAG |
| SEQ_1448 | 1453020 | m6A | CTGCAG |
| SEQ_1597 | 1602240 | m6A | CTGCAG |
| SEQ_1597 | 1602243 | m6A | CTGCAG |
| SEQ_1597 | 1602706 | m6A | CTGCAG |
| SEQ_1615 | 1626084 | m6A | CTGCAG |
| SEQ_1625 | 1634867 | m6A | CTGCAG |
| SEQ_1625 | 1634870 | m6A | CTGCAG |
| SEQ_1627 | 1636655 | m6A | CTGCAG |
| SEQ_1651 | 1658796 | m6A | CTGCAG |
| SEQ_1651 | 1658799 | m6A | CTGCAG |
| SEQ_1895 | 1896398 | m6A | CTGCAG |
| SEQ_1895 | 1896401 | m6A | CTGCAG |
| SEQ_1981 | 1925057 | m6A | CTGCAG |
| SEQ_1981 | 1925060 | m6A | CTGCAG |
| SEQ_1920 | 1928487 | m6A | CTGCAG |
| SEQ_1920 | 1928490 | m6A | CTGCAG |
| SEQ_1937 | 1945433 | m6A | CTGCAG |
| SEQ_1937 | 1945436 | m6A | CTGCAG |
| SEQ_1937 | 1945842 | m6A | CTGCAG |
| SEQ_2009 | 2033472 | m6A | CTGCAG |
| SEQ_2152 | 2161140 | m6A | CTGCAG |
| SEQ_2152 | 2161143 | m6A | CTGCAG |

 

**TABLE 4** (Continued)

| CDS | Location | Type | Motif sequence |
| --- | --- | --- | --- |
| SEQ_2161 | 2171880 | m6A | CTGCAG |
| SEQ_2161 | 2171883 | m6A | CTGCAG |
| SEQ_2161 | 2172181 | m6A | CTGCAG |
| SEQ_2161 | 2172184 | m6A | CTGCAG |
| SEQ_2161 | 2173113 | m6A | CTGCAG |
| SEQ_2161 | 2173116 | m6A | CTGCAG |
| SEQ_2210 | 2224386 | m6A | CTGCAG |
| SEQ_2210 | 2224389 | m6A | CTGCAG |

genetic elements, consistent with previous findings (8). The equibactin locus (*eqbA* to *eqbN*), a novel iron acquisition element, was identified among all the SEE isolates, although the partial or entire deletion of this locus in SEE isolates from the United Kingdom has been reported (8, 28). Interestingly, none of the ICEs or prophages were identified in their entirety among all 50 SEE isolates (Table 1). This finding could be because none of the 50 SEE genomes are fully contiguous, due to more differences in acquired mobile genetic elements in SEE than initially thought, or because the absent portions of these acquired mobile genetic elements are similar to other CDS in the SEZ genomes. Nevertheless, this suggests that there is more variability seen among the CDS found within the ICE and prophages than has been described for SEE 4047. The primary functions described for the 23 CDS from the ClueGO analysis that were unique to SEE were DNA modification and binding, endonuclease activity, ATPase activity, and the KEGG pathways of *Staphylococcus aureus* infection and biosynthesis of siderophore group nonribosomal peptides. The DNA binding, endonuclease activity, ATPase activity, and biosynthesis of siderophore group nonribosomal peptides pathway are all functions related to the equibactin siderophore locus (28). Thus, enhanced iron acquisition might be a mechanism by which SEE is able to survive in the equine host, although it is unclear whether this function is somehow host specific (i.e., enhances iron acquisition specifically or optimally in the equine respiratory tract) or whether genes in the equibactin locus serve functions other than iron acquisition that might confer host specificity. Finally, the superantigens (*seeI*, *seeL*, *seeH*) were a part of the *Staphylococcus aureus* infection pathway, which shares similarities in pathogenesis with SEE and its close relative *S. pyogenes* (30, 34). These superantigens activate multiple T cell populations, and the production

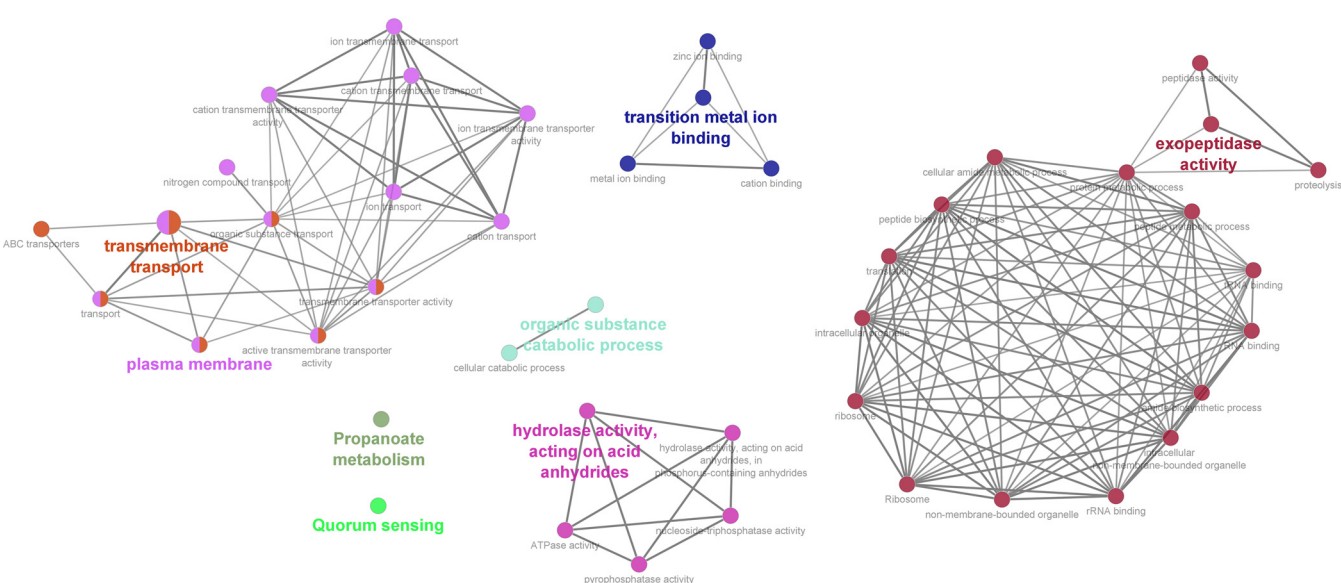

**FIG 6** Gene ontology (GO) terms and KEGG pathways (annotated in ClueGO) on homologous proteins where methylation is present in SEE (*n* = 24) genomes but absent in SEZ (*n* = 24) genomes. Circle size represents the degree of the positive relationship between the GO terms and the term's adjusted *P* value. The related terms are grouped and presented in the same color.

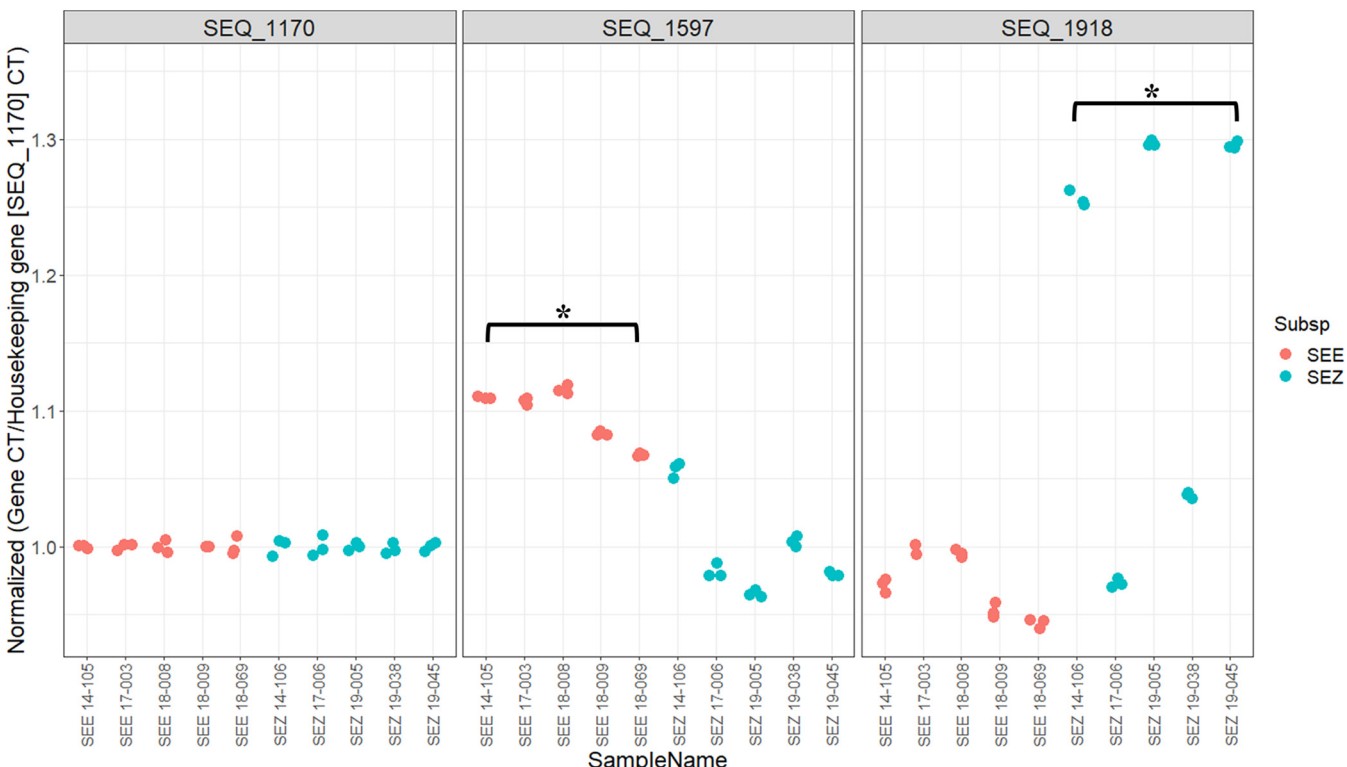

**FIG 7** Reverse transcription $C_T$ values of SEE ($n = 5$) and SEZ ($n = 5$) isolates. $C_T$ values were normalized to the housekeeping gene (SEQ_1170) and demonstrated no difference in transcription level. Normalized $C_T$ values for SEQ_1597 (exopeptidase activity) demonstrated a significantly ($P < 0.05$) lower level of transcription relative to the housekeeping gene in SEE compared to that in SEZ. Normalized $C_T$ values for SEQ_1918 (quorum sensing) revealed significantly ($P < 0.05$) higher levels of transcription relative to the housekeeping gene in SEE compared to those in SEZ. Statistics were performed using the Wilcoxon rank sum test, and an asterisk represents a $P$ value of <0.05. Pink dots denote SEE isolates, and blue dots denote SEZ isolates.

of antibodies can be suppressed through IFN-$\gamma$ production by overactivated CD4 T cells (31, 32). Similarly, the overproduction of the proinflammatory cytokine tumor necrosis factor-$\alpha$ by immune cells that have recognized these superantigens results in suppression of phagocytic cell recruitment to the sites of infection (35). Consequently, these superantigens divert the host's immune responses of antibody-complement opsonization and phagocytic killing of pathogens (36).

Identification of AGEs in all 50 SEZ isolates from the same anatomic location of the same host species from the same geographic region demonstrated the relatively high variability of this bacterial subspecies. Only 15 CDS were identified in all SEZ isolates that were also absent from all the SEE isolates (Table 2). These elements were annotated to functions attributed to fermentation of lactose and sorbitol. Lactose and sorbitol are commonly known to be fermented by SEZ but not by SEE (6), although alternative fermentation profiles have been described (37, 38). Another major difference between SEE and SEZ was in the components of the cytoplasmic membrane, two of which were related to competence stimulation (*comA*, *comB*). However, because of the variability among the genomes of SEZ, it was not possible to identify consistent differences between SEZ and SEE. This variability in the genome of SEZ might explain its ability to adapt to new hosts and environments, whereas SEE might be able to survive better in horses by more efficiently scavenging iron when it is restricted.

The intrasubspecies variances of SEE and SEZ were evaluated by analysis of SNPs that were identified in ≥75% of either the SEE isolates or the SEZ isolates using ParSnp as described previously (39). For SEE, 15 SNPs that occurred in ≥75% of strains were identified, including 3 synonymous mutations (Table S3). Three SNPs were found in the CDS (SEQ_2017) encoding the SeM protein. This finding is not surprising, because variation in the sequence of the SeM protein among SEE strains is common (40). For SEZ, 587 SNPs that occurred in ≥75% of strains were identified. Although SNPs were found

more commonly in SEZ (*n* = 587) than in SEE (*n* = 15) (Table S4), most of the SEZ SNPs (75%; 441/587) were synonymous mutations. Some of the SEZ CDS in which SNPs occurred encoded proteins that were related to cellular components of the membrane, plasma membrane, and cytoplasm, and other CDS were associated with metabolic or biosynthetic processes (Fig. 5, Table S5). The only significant (*P* < 0.05) interaction of the SEZ CDS identified using ClueGO was related to the 2-component signal transduction system which enables bacteria to sense, respond, and adjust to environmental changes (41). This 2-component system has been described to regulate virulence factors such as the capsule of *S. pyogenes* (42). Streptococcal species that possess a Com-type 2-component system are proposed to be readily transformable to their niche (43), and elements of this 2-component system have been previously characterized in a strain of SEZ that caused nephritis (44).

The global methylomes of 24 SEE and 24 SEZ isolates were considered by using PacBio SMRT sequencing and the BaseMod pipeline (21), and sites of methylation on homologous proteins of the 2 subspecies were targeted. We elected to compare methylation of the homologous proteins of SEE and SEZ because of the high degree of similarity in the genomes of SEE and SEZ (6) and to assess if methylation differences might contribute to the host restriction of SEE. The important role of methylation has been described for *S. pyogenes*, the closest relative of SEE and SEZ, wherein the absence of methylation at a prominent motif was demonstrated to alter gene expression that resulted in decreased virulence and altered the bacterium's ability to survive in neutrophils (24). The global methylomes of the 24 SEE isolates were more consistent than those of the 24 SEZ isolates, commensurate with the greater variability of AGEs of the SEZ isolates studied here. Numerous methylation motifs were identified in the SEE and SEZ isolates, including several novel motifs, primarily among the SEZ isolates (Table 3). However, a single novel motif sequence (GGATGNND) was identified in an SEE isolate from Salado, Texas with a methylation frequency of 16%. The motif types that were identified in the SEZ isolates were associated with mostly methylation type m6A, although 1 motif (RGATCY) found in 2 SEZ isolates was the m4C methylation type. While many novel motifs were described, 12 of the 24 SEZ isolates had the motif (CATCC/GGATG) that has been identified in REBASE (Table S6), and this motif was also found in 13 of the 24 SEE isolates. These partnered motifs are both associated with type II restriction modification and methyltransferases in the SEZ H70 genome according to REBASE. Very little is known about the functions of these previously described restriction modification systems. The type II systems have been described as a bacterial immune system by protecting against invasion and modification by foreign DNA or bacteriophage by the presence of methylation at the motif sequence (22, 23, 45). The presence of methylation at each occurrence of this motif sequence (CATCC/GGATG) was much higher (~97%; range 95% to 98%) in the SEE isolates than in the SEZ (~68%; range 48% to 90%). It is possible that this modification reflects a method of adaptation to protect against a bacteriophage that predominates in the respiratory tract of horses that targets SEE or SEZ, whereas the greater diversity of strains of SEZ might enable them to adapt to many hosts or sites for opportunistic infection. It is also possible that this motif sequence (CATCC/GGATG) is an example of changes in methyltransferase activity resulting from acquisition of mobile genetic elements by horizontal gene transfer (22, 46, 47). Although we observed more consistent methylation patterns in SEE isolates, this is likely to be explained in part by the consistency of the acquired mobile genetic elements described for SEE, whereas different strains of SEZ are likely to able to acquire a greater variety of mobile genetic elements, thereby resulting in more variability of methyltransferase activity and methylation patterns.

By selecting genes encoding homologous proteins with methylation present in all SEE (*n* = 24) but absent in all SEZ (*n* = 24) isolates, we identified 37 CDS. All 37 CDS in the SEE genomes had the m6A type modification with the motif sequence CTGCAG (Table 4). REBASE indicates that this modification and methylation motif is from a type II methyltransferase and restriction modification system that has been described in the

Microbiology
Spectrum

strain SEE 4047. The presence of methylation at each occurrence of the motif sequence CTGCAG was highly prevalent (~95%) in all the SEE genomes (Table S6). However, this particular motif sequence was not found among any of the SEZ isolates. The functions of these 37 CDS were assessed using ClueGO, and several GO terms and KEGG pathways were noted (Table S7). Absence or alteration of methylation at motifs has been shown to alter gene expression in several different bacterial species (24–26). Thus, we hypothesize that the absence of methylation at these homologous proteins in SEZ results in altered gene expression of these CDS, resulting in functional differences. Several of the GO functions and KEGG pathways associated with these differentially methylated CDS have not been studied in either of the *Streptococcus equi* subspecies. These noted differences could be important to the pathogenesis and microbe-host interactions for SEE relative to SEZ. The exopeptidase activity was linked to 3 CDS (SEQ_0976, SEQ_1597, SEQ_1920), all of which had GO biological process term associated with peptidase activity (Table S7). Conceivably, this exopeptidase activity could contribute to host specificity or pathogenesis of SEE infection in horses and warrants further investigation. Additionally, many of the differentially methylated genes were linked to proteins that functioned in transmembrane transport and transport of compounds, indicating that whether these functions influence microbe-host interactions specifically in horses warrants further investigation. The quorum-sensing pathway was associated with 3 CDS (SEQ_1918, SEQ_2009, SEQ_0435). Quorum sensing has been described in *S. pyogenes* to play a role in establishing disease in the host and evading the host's immune system (48) and was recently described in SEZ to influence capsule polysaccharide production and biofilm formation (49). Interestingly, a capsule depth for SEE 4047 higher than that for SEZ H70 and considered important in the pathogenesis of strangles has been attributed to an inversion in genes involved in hyaluronate production (6). However, the observed methylation differences could also contribute to the thicker capsule of SEE. Although differences in bacterial function cannot be inferred based on methylation patterns, we believe our results identify targets for further investigation for their role in the host specificity and virulence of SEE.

To assess the functional significance of the observed differential methylation at CDS sites found in SEE but not SEZ (Table S7), we quantified gene expression for 2 representative CDS by qPCR: SEQ_1597 (quorum sensing) and SEQ_1918 (exopeptidase activity). We demonstrated levels of transcription for these genes significantly different from those of a housekeeping gene (SEQ_1170) (Fig. 7). Notably, the levels of transcription for SEQ_1597 decreased in SEE isolates compared to those in SEZ isolates, whereas the level of transcription for SEQ_1918 increased in SEE isolates compared to that in SEZ isolates. This phenomenon has been described in other prokaryotes. The DNA adenine methyltransferase (Dam) of *Salmonella enterica* catalyzes methylation of specific motif sequences (50). Deletion of the gene encoding Dam resulted in either increased or decreased expression of specific genes associated with virulence of *S. enterica* (50). Similarly, in *Borrelia burgdorferi*, deletion of the m6A methyltransferase genes resulted in the absence of methylation of motif sequences that caused increased or decreased expression of genes associated with the ability to colonize the host and with vector acquisition/transmission (26). However, the absence of methylation at a motif sequence in *S. pyogenes* resulted in downregulation of genes associated with decreased virulence and reduced ability to replicate within neutrophils (24). Although we evaluated only 2 CDS, our results indicate that studying the global methylome could shed further light on the phenotypic differences between these *Streptococcus* subspecies and that these differentially methylated sites in SEE and SEZ (i.e., methylation present in SEE but absent in SEZ) could play a role in the host specificity of SEE.

Comparisons of methylation sites on homologous proteins in the 24 SEZ isolates that were absent in the 24 SEE isolates demonstrated a degree of variability greater than that observed for those present in SEE but absent in SEZ. The methylation summaries of the SEZ isolates yielded far more inconsistent methylation motif sequences, and 3 SEZ isolates lacked specific methylation motif sequences (19-005, 19-051, 19-

053) (Table S6). This variability in methylation might contribute to the ability of SEZ to infect or colonize a wider number of hosts (13–17), whereas a far more restricted methylation repertoire was found in the single-host pathogen SEE. Initially, 10 potential CDS were identified where methylation was present in the SEZ isolates but absent on the homologous protein in the SEE isolates. However, upon further investigation, it was determined that the presence of methylation did not occur at the same location within the CDS, did not have the same motif sequence, and sometimes differed in the type of methylation present (m6A or m4C) (Table S8). Therefore, it is difficult to draw conclusions about the impact of the presence of methylation at these 10 homologous proteins. Collectively, our methylome results that CDS methylated in SEE but not in SEZ were associated with altered gene expression suggest that the differential methylation in SEE contributes to the host specificity of SEE. A more comprehensive analysis of the differentially methylated CDS in SEE and SEZ is warranted.

This study has several limitations. The first limitation of this study is the incomplete genomes of the SEE and SEZ isolates. Although the use of PacBio sequencing allows for more contiguous draft genome (i.e., fewer number of contigs, range 1 to 59) than using short-read technologies, gaps in the genome remained (Table S10). The PacBio SMRT sequencing, however, enabled us to study the complete methylomes of these bacteria. Another limitation of our study was the necessity to utilize reference genomes for the characterization of the AGEs and methylation patterns of both SEE and SEZ genomes. The reference genome selected creates bias, and this was especially apparent for SEZ where we identified marked variability of the genomic elements both in the accessory genome and in the methylation patterns. However, the information derived from these 50 SEZ genomes and their methylation pattern will increase the publicly available genomic data. An important limitation was that we assessed the association of differential methylation of the homologous proteins in SEE and SEZ isolates with a difference in a functional response (*viz.*, gene expression) for only 2 CDS. Unfortunately, further evaluation of differential gene expression was beyond the scope of available funding. Further evaluation of the functional impact of the differential methylation is warranted. Despite this limitation, to the authors' knowledge, this report is the first characterization and comparison of the global methylomes in SEE and SEZ isolates from the United States.

In this study, we described the differences in the accessory genome (i.e., elements that are not present in all isolates of the bacterial subspecies) and complete methylation patterns of SEE and SEZ isolates from Texas. We described that the majority of AGEs found in all 50 SEE isolates were attributed to the mobile genetic elements (ICE and prophages) described in the reference SEE 4047. Fewer AGEs were found in all 50 SEZ isolates or involved in lactose and sorbitol fermentation, but we also identified genes related to competence stimulation that were not identified in SEE. Global methylomes were characterized for 24 SEE and 24 SEZ isolates, and we noted more consistent patterns of methylation in the SEE isolates than in the SEZ isolates. We identified 19 novel methylation motifs primarily among the SEZ isolates. Importantly, we identified methylation of homologous proteins present in SEE but absent in SEZ and found evidence that differential methylation for some of these genes was associated with altered gene expression. A more comprehensive evaluation of all these proteins that are methylated in SEE but not SEZ is warranted to investigate whether they might be candidates for explaining the mechanism(s) of the host specificity and pathogenesis of SEE. Finally, we were unable to consistently identify sites of methylation present in SEZ but absent in SEE on homologous proteins, which lends further credence to the possibility that the differentially methylated genes in SEE contribute to host specificity. Much remains to be learned about the impact of methylation on the differences in SEE and SEZ. In summary, the finding that comparison of the genomes and methylomes did not readily identify differences that explain the host specificity of SEE indicates that it will be necessary to evaluate host-microbe interactions to unravel what drives specificity of SEE for infecting horses, using both *in vitro* and *in vivo* systems.

## MATERIALS AND METHODS

***Streptococcus equi* isolates.** Fifty (50) SEE isolates and 50 SEZ isolates were selected to be included in this study (Table S9). The SEE isolates were collected from horses from various regions of Texas during multiple years (2012 to 2019), aiming for a more representative and geographically diverse population of isolates because the referral base for our teaching hospital and our state veterinary diagnostic laboratory are based largely in central Texas (Fig. S1). The 50 SEZ isolates were selected from the respiratory tract of horses from various regions of Texas, from multiple years (2010 to 2020), and were representative of the differing disease states recognized for SEZ in horses (i.e., commensal and virulent isolates).

**Bacterial DNA extraction and whole-genome sequencing.** The *Streptococcus* isolates were cultured for 24 h in 3 mL of Todd Hewitt (TH) medium (HIMEDIA, West Chester, PA, USA) in 5% $CO_2$ at 37℃. Following a 24-h incubation, the isolates were centrifuged at 3,000 $\times$ *g* for 10 min to create a pellet. The supernatants were discarded, and DNA extractions were performed using the DNeasy UltraClean microbial kit (Qiagen, Hilden, Germany), following the manufacturers' instructions with slight modifications. Briefly, the bacteria pellets were resuspended in 300 $\mu$L of PowerBead solution and transferred into PowerBead tubes. Fifty microliters (50 $\mu$L) of solution SL was added, and the PowerBead tubes were incubated at 70℃ for 10 min, followed by horizontal vortexing for an additional 10 min. Then, the PowerBead tubes were centrifuged at 10,000 $\times$ *g* for 30 s and the supernatants were transferred to new tubes. One hundred microliters (100 $\mu$L) of solution IRS was added to the supernatants, incubated for 15 min at 4℃, and then centrifuged at 10,000 $\times$ *g* for 1 min. The supernatants were transferred to new tubes without disturbing the pellet, and 900 $\mu$L of solution SB was added and mixed thoroughly. Seven hundred microliters (700 $\mu$L) of this solution was transferred to MB spin column tubes and centrifuged at 10,000 $\times$ *g* for 30 s, and the flowthrough was discarded, and then this step was repeated. Additionally, 300 $\mu$L of solution CB was added to the columns and centrifuged at 10,000 $\times$ *g* for 30 s. Then, another centrifugation step (10,000 $\times$ *g* for 1 min) was performed to remove any excess fluid and the MB spin columns were transferred to new collection tubes. Finally, 50 $\mu$L of the solution EB was added to the columns and centrifuged at 10,000 $\times$ *g* for 30 s. The DNA quality and concentrations were measured using the NanoDrop spectrophotometer (ND-1000, Thermo Fisher Scientific, Waltham, MA, USA) and sent to the Duke Center for Genomic and Computational Biology (GCB) for WGS on the PacBio Sequel platform.

**Bioinformatic analysis.** After the completion of WGS at GCB, raw subreads were assembled into genomes *de novo* using CANU (v7.0) (51) on the Texas A&M High Performance Research Computing cluster. The assembled genomes' length and number of contigs were noted (Table S10), and genomes were confirmed to be SEE or SEZ through ribosomal MLST (52). The genomes were then annotated with RASTtk (v2.0) (53), using the web-based server. Following annotation, the genomes were input into Spine (v0.3.2) (19) to define the core genome (i.e., elements found in all genomes) of both *Streptococcus equi* subspecies. Using the core genome output from Spine, the accessory genomes (i.e., elements present in some genomes but absent from others) for each isolate were identified using AGEnt (v0.3.1) (19). Finally, ClustAGE (v0.8) (54) was used to identify and group the AGEs into bins for the SEE and SEZ genomes. The graphical representation of bins with clustered AGEs by each individual genome was performed with the ClustAGE plot (http://vfsmspineagent.fsm.northwestern.edu/cgi-bin/clustage_plot.cgi). Using a custom R script (v4.0.3) (Appendix) (55), bins were identified with AGEs specific to either all SEE (*n* = 50) or all SEZ (*n* = 50). The genes of the AGEs within the selected bins with ≥95% of the protein identified were included and were compared to their respective reference genomes (SEE 4047 or SEZ H70). Using the Cytoscape (v.3.8.2) (56) plug-in ClueGO (v2.5.7) (57), the Gene Ontology (GO) terms and pathway interactions for the AGEs of SEE and SEZ were evaluated using default parameters, and the localization of the protein within the cell was determined using PSORTb (v3.0) (58).

Using the CANU-assembled genomes, the SNPs of the core genomes for both SEE and SEZ were also considered using ParSnp (59) as described previously (39). Briefly, SEE or SEZ genomes were aligned by their core genomes against the appropriate reference genome (either SEE 4047 or SEZ H70) using ParSnp. The ParSnp output was converted to a variant call format (VCF) file and the percentage of variance was determined by summing the total number of variants in the VCF file for each genome. The determined totals for each genome were then divided by the length of the ParSnp-defined core genome, and graphs were generated using ggplot2 (v3.3.5) (60). Using the Cytoscape (v.3.8.2) (56) plug-in and ClueGO (v2.5.7) (57), the Gene Ontology (GO) terms and KEGG pathway interactions for the SNPs of SEZ were evaluated using default parameters.

The complete methylation profiles of a subset of SEE (*n* = 24) and SEZ (*n* = 24) genomes were characterized; these isolates were selected to be representative of distribution across the phylogenetic tree (Fig. S2 to S4). The phylogenetic trees were built using the PATRIC (61) phylogenetic tree building service. Phylogenetic tree outputs from PATRIC were viewed and edited using Microreact (v5.123.1) (62). The complete methylomes were characterized with the BaseMod (https://github.com/ben-lerch/BaseMod-3.0) pipeline in the PacBio SMRT Link (v8.0) command line tools. Briefly, pbmm2 was used to align the raw BAM files to the appropriate reference genome (i.e., SEE 4047 or SEZ H70). Using the aligned BAM files, the kineticTools function "ipdSummary" was implemented to generate GFF and CSV files with the base modification information. Next, the MotifMaker "find" function was used to generate a second set of CSV files that identified consensus motifs. Finally, the execution of the MotifMaker "reprocess" function generated GFF files with all of the modifications that were part of the motifs. Using R (v4.0.3), the motif GFF files were filtered based on having the presence of a known methylation type (m4C or m6A) and having a QV score (i.e., a quality measure of the detection event) of ≥30. These filtered GFF files of SEE or SEZ genomes were then annotated by either the SEE 4047 or SEZ H70 reference genome, respectively, using the BedTools (63) "annotate" function. Annotated outputs were then compared across the SEE and SEZ genomes for the presence or absence of methylation of homologous proteins using custom scripts in R (Appendix). A list of homologous proteins (≥99% identity) from SEE 4047 and SEZ H70 was generated using the PATRIC

**TABLE 5** Real-time PCR primer sequences

| Gene ID | Target function | Forward primer | Probe | Reverse primer |
| --- | --- | --- | --- | --- |
| SEQ_1170 | Housekeeping | ATACGATGACCTACTGGCTTTG | FAM-ATGATGGCTTCGATACGCTCTGGC-NFQ | CGCTTGATCTCCTCCATTTCT |
| SEQ_1597 | Exopeptidase | GCTGGCAATGACCTGATTCT | FAM-ATGTGCTTCTCAGTTGAGCCTGGT-NFQ | CGTGACCACAGTCCTCAATAC |
| SEQ_1918 | Quorum sensing | GGCAGCACATAAGACCTAACA | FAM-TAGCAGCGGAAAGGTATCTGGCAA-NFQ | GCCAAGCATTGCGCTTATC |

proteome comparison. Identified motifs were then compared to the SEE 4047 and SEZ H70 genomes using REBASE (29). The Cytoscape (v.3.8.2) (56) plug-in ClueGO (v2.5.7) (57) was implemented using default parameters to assess the GO terms and pathway interactions for the different sites of methylation among the SEE and SEZ genomes. The Linux and R codes for this work are provided in the supplemental material (Appendix).

**RNA extraction and qPCR.** SEE ($n = 5$) and SEZ ($n = 5$) isolates were grown in TH medium for 4 h (exponential-phase growth) at 37°C in 5% $CO_2$. Following the 4-h incubation, liquid cultures were centrifuged at 3,000 × *g* for 10 min to pellet the bacterium and the supernatants were discarded. The bacterial RNAs were then extracted using the RiboPure RNA purification kit (Ambion RiboPure-bacteria kit; Invitrogen, Carlsbad, CA, USA) following the manufacturer's instructions. Briefly, the SEE and SEZ pellets were resuspended in 350 $\mu$L of the RNA$_{WIZ}$ solution and then transferred to tubes with Zirconia beads. The tubes were placed on a horizontal vortex adaptor, beat for 10 min at maximum speed, and then centrifuged at 13,000 × *g* for 5 min at 4°C. The supernatants containing the lysed bacteria were transferred to fresh tubes, 0.2 volumes of chloroform were added, and samples were incubated for 10 min at 22°C. To separate the organic and aqueous phases, tubes were centrifuged at 13,000 × *g* for 5 min at 4°C. The aqueous phases were transferred to new tubes, and 0.5 volumes of 100% ethanol were added, mixed thoroughly, and transferred to filter cartridges in 2-mL tubes. The filter cartridge tubes were then centrifuged at 13,000 × *g* for 1 min, the flowthrough was discarded, and the filters were washed by the addition of 700 $\mu$L of wash solution 1. A second and a third wash step were performed with wash solution 2/3. After the third wash step, the filter cartridges were transferred to new tubes. Finally, the RNA was eluted by 50 $\mu$L of elution solution and a DNase treatment was performed. The quality and purity of the RNAs were assessed using the NanoDrop (ND-1000, Thermo Fisher Scientific, Waltham, MA, USA).

Following the RNA extraction, transcription of the genes SEQ_1597 (putative Xaa-Pro dipeptidase), SEQ_1918 (putative oligopeptide transporter permease protein), and housekeeping gene SEQ_1170 (*gyrA*) (6, 8) were quantified using qPCR. Primer and probe sequences were generated using NCBI Primer BLAST (64) and PrimerQuest Tool (https://www.idtdna.com/Primerquest/Home/Index) (Table 5). Reverse transcription and transcript quantification were performed using the TaqMan fast virus 1-step master mix (Thermo Fisher Scientific, Waltham, MA, USA) following the manufacturer's recommendations. Each reaction contained 5 $\mu$L of master mix, 1 $\mu$L of Custom TaqMan gene expression assay (20×, Thermo Fisher Scientific, Waltham, MA, USA), 1 $\mu$L of bacterial RNA (up to 100 ng/$\mu$L), and double-distilled water (dd$H_2$O) made up to a final volume of 20 $\mu$L. These reactions were thermo-cycled for 5 min at 50°C, then for 20 s at 95°C, followed by 40 cycles of 3 s at 95°C and 30 s at 60°C using the QuantStudio 6 Flex (Thermo Fisher Scientific, Waltham, MA, USA). $C_T$ values of SEQ_1597 and SEQ_1918 were normalized to the mean $C_T$ value of the housekeeping gene (SEQ_1170). Statistical significance of normalized $C_T$ values was assessed using the Wilcoxon rank sum test in R (v4.1.0), and significance was set at $P < 0.05$.

**Data availability.** WGS and BAM files were submitted to NCBI's GenBank and Sequence Read Archive (SRA), respectively, under the BioProject number PRJNA763470. For specific genome accession numbers, please see Table S10 in the supplemental material.

## SUPPLEMENTAL MATERIAL

Supplemental material is available online only.

**SUPPLEMENTAL FILE 1**, XLSX file, 0.2 MB.
**SUPPLEMENTAL FILE 2**, PDF file, 0.4 MB.

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
