## [Reviewer comments · Microbiology Spectrum]

Microbiology Spectrum

Differences in the accessory genomes and methylomes of strains of *Streptococcus equi* subsp. *equi* and of *Streptococcus equi* subsp. *zooepidemicus* obtained from the respiratory tract of horses from Texas

Ellen Ruth Morris, Jing Wu, Angela Bordin, Sara Lawhon, and Noah Cohen

Corresponding Author(s): Noah Cohen, Texas A&M University

Review Timeline:

Submission Date:	June 30, 2021
Editorial Decision:	August 17, 2021
Revision Received:	October 14, 2021
Editorial Decision:	December 5, 2021
Revision Received:	December 10, 2021
Accepted:	December 10, 2021

Editor: Cheryl Andam

Reviewer(s): The reviewers have opted to remain anonymous.

Transaction Report:

DOI: <https://doi.org/10.1128/Spectrum.00764-21>

August 17, 2021

Dr. Noah D Cohen
Texas A&M University
Large Animal Medicine & Surgery
College of Veterinary Medicine
Texas A&M University
College Station, TX 77843-4475

Re: Spectrum00764-21 (Differences in the accessory genomes and methylomes of strains of *Streptococcus equi* subsp. *equi* and of *Streptococcus equi* subsp. *zooepidemicus* obtained from the respiratory tract of horses from Texas)

Dear Dr. Noah D Cohen:

While the overall evaluation of the manuscript is generally positive, the reviewers have brought up important concerns. Therefore, I invite you to respond to the referees' comments and revise your manuscript, but please consider the comments carefully and take them into account during the revision process.

Thank you for submitting your manuscript to Microbiology Spectrum. When submitting the revised version of your paper, please provide (1) point-by-point responses to the issues raised by the reviewers as file type "Response to Reviewers," not in your cover letter, and (2) a PDF file that indicates the changes from the original submission (by highlighting or underlining the changes) as file type "Marked Up Manuscript - For Review Only". Please use this link to submit your revised manuscript - we strongly recommend that you submit your paper within the next 60 days or reach out to me. Detailed information on submitting your revised paper are below.

Link Not Available

Sincerely,

Cheryl Andam

Journals Department
Reviewer comments:

Reviewer #1 (Comments for the Author):

The authors present a competently performed study and well written manuscript detailing the WGS of 50 isolates each of *Streptococcus equi* ssp. *equi* and *Streptococcus* ssp. *zooepidemicus* from Texas. Comparative genomic analysis of these related organisms are rather limited despite the potential that they have to give insight into the evolution and disease pathogenesis of the important equine pathogen *Streptococcus equi*. The main focus of the work are differences between the two subspecies in terms of their accessory genome and genome methylation patterns. The work is of high technical quality and the genomes and analysis are a useful contribution to this area of study.

Specific comments (all relatively minor) are:

1. Line 22 underlined
2. Line 50 'that' repeated

3. Lines 33 and 50 I feel would be better qualified with the phrase 'may contribute' given the current observational evidence.
4. Lines 148 and 279 should 'proteins' not be 'genes' given it is the genome and its methylation that is being investigated here?
5. Line 234 suggest 'comA, comB' as per convention for gene names.
6. Table S6 can TTW, GP lavage and the collection source be defined?
7. How where the phylogenetic trees in Figures S1-3 constructed?
8. Tables 1 and 2, gene names should be italicised.
9. Table S3 most of the columns for isolates 19-005, 19-051 and 19-053 are empty, is this data complete?

Reviewer #3 (Comments for the Author):

This paper compares the genomes and methylomes of 50 SEE and SEZ isolates to further understand the evolution of SEE an equine specific pathogen. The manuscript primarily focuses on shared genes and methylation sites/patterns found in SEE that are not present in SEZ and visa versa.

Overall, the writing needs to be tightened throughout the document for increased clarity and succinctness.

The characterization of the methylomes is interesting but the genome comparisons feel incomplete.

Could you expand your analyses for example to account for SNPs?

Can you give an actually estimate of the differences in intra-species diversity among the SEE and SEZ isolates as well as inter-species diversity? This would certainly help illustrate your point that these populations have different characteristics. For example, the average diversity among SEE genomes was X% while the average diversity of SEZ genomes was X%.

Can you talk about gene duplications, which are supposed to be a major source of evolutionary adaptation for SEE?

Can you include a more complete summary in the results regarding how complete these genomes are? Are the gene losses you identified

really just a case of incomplete genomes? It's difficult to asses as there are no quality metrics given regarding your assemblies.

Quality metrics could be given in a supplementary Table to assist the reader.

One of the major issues I have with this manuscript is the framing of ISs as key to the evolution of host restriction.

If SEE has diverged from an ancestral SEZ bacterium, it's likely that SEE underwent a bottleneck effect (as you cite). This would constrain its

genetic diversity and likely the number of species it could infect. Gene loss seems a more probable cause for host restriction.

From my cursory reading of other papers on SEE genomes, the proposal is that gene loss and duplication are mediated by ISs, which could limit SEE to a single host. It's not the presence of ISs and mobile genetic elements alone that cause the constraint. I feel like

the gist of this hypothesis has been mis-summarized. There is also the claim of a key siderophore in SEE being responsible for host-specificity, which seems

touched upon in the discussion. It would also be nice for the reader to have some understanding of the geographical distribution of these isolates - maybe a supplemental figure of isolation sites in Texas?

Line 27 - do you mean fewer shared genome elements?

"Fewer accessory genome elements were identified in SEZ because of the greater genomic variability observed among these isolates."

Technically, a gene in one SEZ that is not present in the others would

be classified as an accessory gene. To me accessory genes describe those within a species not those between two species or subspecies

so the way that this term is used is a bit unorthodox. I think it would be in-line with standard practices to talk about accessory genes within

the pan-genome of SEE or SEZ but not between them. A comparison between SEE and SEZ would reveal orthologous sequences as well as unique genes.

Line 30 - this sentence is confusing do you mean that shared methylation patterns were only found among homologous genes of SEE?

Also change homologous proteins to genes.

"Among homologous proteins of SEE and SEZ, the presence of methylation was only identified in genes of SEE isolates encoding proteins with functions"

Could some enrichment analysis be performed to supplement the information provided in Figures 1-3?

Figure S2

It's unclear how genomes were chosen for diversity according to this phylogeny. Most genomes are essentially identical.

Why were so many identical genomes chosen (for example, the four yellow genomes at the top of the tree)

over those that sit on longer branches, which indicate more change from a common ancestor.

Harris et al has sequenced the genomes of several hundred SEE from global locations, how do these compare to your isolates? I would expect the geographic constraint of your research to influence the diversity observed. Can you comment on this?

Examples of sentences that should be reworded for clarity and succinctness

Please reword the sentences in the discussion as ATPase activity, DNA modification, endonuclease activity, etc are certainly not functions specific to SEE.(starting on Line 212)

The 23 coding sequences might be unique to SEE and represent a unique pathway but these are not unique functions: The primary functions described for the 23 CDS from the ClueGO analysis were DNA modification and binding, endonuclease activity, ATPase activity, and the KEGG pathways of Staphylococcus aureus infection and biosynthesis of siderophore group nonribosomal peptides. Thus, these reflect functions specific to SEE.

Line 241:

whereas SEE might have evolved to more specifically infect horses (possibly by more efficiently scavenging iron when it is restricted).

Please change this statement as it suggests some sort of deterministic effort by SEE to evolve.

SEE specifically infects horses due to evolutionary constraints, stochastic processes, selective pressures, etc.

Line 71

Please reword this sentence.

Although SEE has relatively reduced genetic diversity, greater genetic variation has been described for isolates of SEZ (14, 18) For example, "Greater genetic variation is described for isolates of SEZ than SEE"

Line 98

change:

evolved to a host-specific pathogen.

to

evolved to become a host-specific pathogen.

Line 101

This sentence is somewhat redundant. Change:

Comparisons of the accessory genome of the 50 SEE and 50 SEZ isolates were performed using the Spine, AGEnt, and ClustAGE pipeline (Fig. 1) to generate the AGEs identified among these isolates (Fig. 1).

To

Comparisons of the accessory genome of the 50 SEE and 50 SEZ isolates were performed using the Spine, AGEnt, and ClustAGE pipeline.

Line 109

What is SEQ_1102, why is it relevant in the context of this paragraph?

Line 111

This wording is confusing. Please tighten.

Interestingly, all of the CDS that form each of the described ICE or prophage from SEE 4047 were not found in all our 50 SEE isolates

change to:

"The CDSs associated with the ICE and prophages from SEE 4047 were not common to all 50 SEE isolates."

Line 114, you introduce the equibactin locus without informing the reader of what this is or why it's relevant. Some context here would

be appreciated. I don't know what SEQ_1233 - SEQ_1246 are.

Line 118

change:

and 23 CDS were characterized

to:

which, characterized 23 CDS.

Line 126

Tighten this sentence.

Change to:

"Twelve of the fifteen SEZ-specific AGEs were localized to the cytoplasm (n = 5) or the cytoplasmic membrane (n = 7) in the bacterium, and single hypothetical protein was predicted to have an extracellular function."

Line 131, these sentences could be combined to be more succinct and less redundant.

The functions of the 15 CDS were evaluated using ClueGO, and a function was identified for only 3 CDS (Fig. 3, Table S2). Unsurprisingly, galactose metabolism was the only GO term described from the 3 CDS (lacE, lacF, and lacG).

Change to

"ClueGo evaluated the function of the 15 CDS, which characterized SZO_15220 - SZO_15250 and SZO_01750 as associated with galactose metabolism."

Staff Comments:

Preparing Revision Guidelines

For complete guidelines on revision requirements, please see the Instructions to Authors at [link to page]. **Submissions of a paper that does not conform to Microbiology Spectrum guidelines will delay acceptance of your manuscript.**

Please return the manuscript within 60 days; if you cannot complete the modification within this time period, please contact me. If you do not wish to modify the manuscript and prefer to submit it to another journal, please notify me of your decision immediately so that the manuscript may be formally withdrawn from consideration by Microbiology Spectrum.

If you would like to submit an image for consideration as the Featured Image for an issue, please contact Spectrum staff.

September 9, 2021

Professor Cheryl Andam
Editor, *Microbiology Spectrum*

Re: Spectrum00764-21 (Differences in the accessory genomes and methylomes of strains of *Streptococcus equi* subsp. *equi* and of *Streptococcus equi* subsp. *zooepidemicus* obtained from the respiratory tract of horses from Texas)

Dear Prof. Andam:

Thank you for your electronic message dated August 17, 2021 pertaining to the above-referenced manuscript. We thank the reviewers for their careful consideration of our report. We have revised the manuscript on the basis of their insightful suggestions and uploaded the revised document with changes to the manuscript tracked. Here we provide a point-by-point response to each comment from each reviewer. Thank you and the reviewers for considering our responses and our revised report.

Comments from Reviewer 1

The authors present a competently performed study and well written manuscript detailing the WGS of 50 isolates each of *Streptococcus equi* ssp. *equi* and *Streptococcus* ssp. *zooepidemicus* from Texas. Comparative genomic analysis of these related organisms are rather limited despite the potential that they have to give insight into the evolution and disease pathogenesis of the important equine pathogen *Streptococcus equi*. The main focus of the work are differences between the two subspecies in terms of their accessory genome and genome methylation patterns. The work is of high technical quality and the genomes and analysis are a useful contribution to this area of study.

AUTHORS' RESPONSE: We thank the reviewer for their kind remarks.

1. Line 22 underlined

AUTHORS' RESPONSE: We have removed the underline from line 22. Our intent had been to emphasize that we compared multiple strains.

2. Line 50 'that' repeated

AUTHORS' RESPONSE: We apologize for and have corrected this mistake.

3. Lines 33 and 50 I feel would be better qualified with the phrase 'may contribute' given the current observational evidence.

AUTHORS' RESPONSE: The manuscript was revised as suggested.

4. Lines 148 and 279 should 'proteins' not be 'genes' given it is the genome and its methylation that is being investigated here?

AUTHORS' RESPONSE: We think the reviewer is correct and have revised the manuscript accordingly.

5. Line 234 suggest 'comA, comB' as per convention for gene names.

AUTHORS' RESPONSE: We apologize for this error and have corrected the manuscript.

6. Table S6 can TTW, GP lavage and the collection source be defined?

AUTHORS' RESPONSE: We have revised Table S6 to spell out TTW and GP before their first use, and to correct some other inconsistencies in format. We regret not being more attentive to these details.

7. How where the phylogenetic trees in Figures S1-3 constructed?

AUTHORS' RESPONSE: The manuscript has been revised to indicate how Figures S1 and S3 were constructed.

8. Tables 1 and 2, gene names should be italicised.

AUTHORS' RESPONSE: We apologize for this error and have corrected Tables 1 and 2 as suggested.

9. Table S3 most of the columns for isolates 19-005, 19-051 and 19-053 are empty, is this data complete?

AUTHORS' RESPONSE: We have revised Table S3 to indicate that NA means not available and filled the rows for these 3 isolates with NA's to indicate the data are not available.

Comments from Reviewer 3

This paper compares the genomes and methylomes of 50 SEE and SEZ isolates to further understand the evolution of SEE an equine specific pathogen. The manuscript primarily focuses on shared genes and methylation sites/patterns found in SEE that are not present in SEZ and visa versa.

Overall, the writing needs to be tightened throughout the document for increased clarity and succinctness.

AUTHORS' RESPONSE: We have strived to be clearer and have revised the manuscript on the basis of specific suggestions from the reviewer detailed below.

The characterization of the methylomes is interesting but the genome comparisons feel incomplete.

AUTHORS' RESPONSE: We thank the reviewer for their helpful suggestions and have responded below to specific comments and recommendations from the reviewer.

Could you expand your analyses for example to account for SNPs?

AUTHORS' RESPONSE: The purpose of our report was to identify genes that differed between SEE and SEZ that might allow us to better understand which genes might reflect host-restriction of SEE. Thus, we did not include analysis across the genomes of these organisms. Although this was not part of our original plan for analysis, we have expanded the manuscript to include an analysis of intraspecies SNPs.

Can you give an actual estimate of the differences in intra-species diversity among the SEE and SEZ isolates as well as inter-species diversity? This would certainly help illustrate your point that these populations have different characteristics. For example, the average diversity among SEE genomes was X% while the average diversity of SEZ genomes was X%.

AUTHORS' RESPONSE: We thank the reviewer for this helpful suggestion. The manuscript has been revised to provide estimates of the intra-subspecies diversity of SEE and SEZ.

Can you talk about gene duplications, which are supposed to be a major source of evolutionary adaptation for SEE?

AUTHORS' RESPONSE: We appreciate the reviewer's point but our specific aims were to compare the accessory genomes and methylomes of these isolates, and analysis of gene duplications was not included in our analysis. Our sequence data will be available to other investigators for pursuing this and other ideas.

Can you include a more complete summary in the results regarding how complete these genomes are? Are the gene losses you identified really just a case of incomplete genomes? It's difficult to assess as there are no quality metrics given regarding your assemblies. Quality metrics could be given in a supplementary Table to assist the reader.

AUTHORS' RESPONSE: We thank the reviewer for this helpful suggestion. We have provided quality metrics in Supplementary Table 7 to assist readers with interpreting the data.

One of the major issues I have with this manuscript is the framing of ISs as key to the evolution of host restriction. If SEE has diverged from an ancestral SEZ bacterium, it's likely that SEE underwent a bottleneck effect (as you cite). This would constrain its genetic diversity and likely the number of species it could infect. Gene loss seems a more probable cause for host restriction. From my cursory reading of other papers on SEE genomes, the proposal is that gene loss and

duplication are mediated by ISs, which could limit SEE to a single host. It's not the presence of ISs and mobile genetic elements alone that cause the constraint. I feel like the gist of this hypothesis has been mis-summarized. There is also the claim of a key siderophore in SEE being responsible for host-specificity, which seems touched upon in the discussion.

AUTHORS' RESPONSE: We thank the reviewer for this very helpful suggestion. We are unsure where we have misrepresented findings about the role of mobile genetic elements, but we think the reviewer might be referring to the first paragraph of the Discussion where we consider the functions encoded by the mobile genetic elements. While we understand the reviewer's point, it seems appropriate and relevant to discuss the functions associated with the products of the mobile genetic elements identified as part of the AGEs and how these might contribute to host-specificity and pathogenesis in ways other than mediating reduced genetic diversity through gene losses and duplications. As noted, this would include the equibactin locus which encodes an iron-acquisition element (siderophore) that might somehow contribute to host-restriction, which is mentioned in our discussion.

It would also be nice for the reader to have some understanding of the geographical distribution of these isolates - maybe a supplemental figure of isolation sites in Texas?

AUTHORS' RESPONSE: We have revised the manuscript to include maps for the geographical distribution of isolates from Texas included in the study. The distribution reflects the referral base for our teaching hospital and state veterinary diagnostic laboratory (both in College Station in central Texas) and the more densely populated regions of Texas.

Line 27 - do you mean fewer shared genome elements? "Fewer accessory genome elements were identified in SEZ because of the greater genomic variability observed among these isolates." Technically, a gene in one SEZ that is not present in the others would be classified as an accessory gene. To me accessory genes describe those within a species not those between two species or subspecies so the way that this term is used is a bit unorthodox. I think it would be in-line with standard practices to talk about accessory genes within the pan-genome of SEE or SEZ but not between them. A comparison between SEE and SEZ would reveal orthologous sequences as well as unique genes.

AUTHORS' RESPONSE: We mean that fewer genomic elements present in some but absent in other isolates of SEZ were identified. We have revised the abstract to try to make this more clear. The approach we have used for defining accessory genome elements has been used to characterize bacteria both between and within species (please see references 19 and 20).

Line 30 - this sentence is confusing do you mean that shared methylation patterns were only found among homologous genes of SEE? Also change homologous proteins to genes. "Among

homologous proteins of SEE and SEZ, the presence of methylation was only identified in genes of SEE isolates encoding proteins with functions"

AUTHORS' RESPONSE: We have revised the abstract to try to make our point clearer, and readers will be able to find more detailed description in the manuscript. We welcome suggestions to make this clearer.

Could some enrichment analysis be performed to supplement the information provided in Figures 1-3?

AUTHORS' RESPONSE: Respectfully, we have provided a number of additional analyses and believe that our revised manuscript adequately addresses our aims without this addition.

Figure S2: It's unclear how genomes were chosen for diversity according to this phylogeny. Most genomes are essentially identical. Why were so many identical genomes chosen (for example, the four yellow genomes at the top of the tree) over those that sit on longer branches, which indicate more change from a common ancestor.

AUTHORS' RESPONSE: We thank the reviewer for this feedback and we will keep this point in mind for future work. The genomes were selected based on placement in the phylogenetic tree to the best of our ability. However, some files were not compatible with the pipeline used for the methylation analysis such that we selected some identical genomes.

Harris et al has sequenced the genomes of several hundred SEE from global locations, how do these compare to your isolates? I would expect the geographic constraint of your research to influence the diversity observed. Can you comment on this?

AUTHORS' RESPONSE: Respectfully, these questions are beyond the scope of our stated aims and objectives. We do expect the geographic restriction to constrain the diversity of our isolates, and we intentionally restricted ourselves to isolates from Texas and from the respiratory tract of horses to maximize chances that differences we observed weren't confounded by geographical differences. Our isolates will be publicly available such that investigators can make comparisons among regions. Such comparisons are important but beyond the scope of our project.

Please reword the sentences in the discussion as ATPase activity, DNA modification, endonuclease activity, etc are certainly not functions specific to SEE.(starting on Line 212). The 23 coding sequences might be unique to SEE and represent a unique pathway but these are not unique functions: The primary functions described for the 23 CDS from the ClueGO analysis were DNA modification and binding, endonuclease activity, ATPase activity, and the KEGG pathways of Staphylococcus aureus infection and biosynthesis of siderophore group nonribosomal peptides. Thus, these reflect functions specific to SEE.

AUTHORS' RESPONSE: The reviewer's point is a good one. We have revised the manuscript to try to improve it by deleting the sentence about these reflecting functions specific to SEE and indicating that the 23 CDS from ClueGO analysis were unique to SEE.

Line 241: whereas SEE might have evolved to more specifically infect horses (possibly by more efficiently scavenging iron when it is restricted). Please change this statement as it suggests some sort of deterministic effort by SEE to evolve. SEE specifically infects horses due to evolutionary constraints, stochastic processes, selective pressures, etc.

AUTHORS' RESPONSE: We thank the reviewer for this suggestion and have revised the sentence as follows: "... whereas SEE might be able to better survive in horses as a result of more efficient scavenging of iron when it is restricted."

Line 71: Please reword this sentence. Although SEE has relatively reduced genetic diversity, greater genetic variation has been described for isolates of SEZ (14, 18) For example, "Greater genetic variation is described for isolates of SEZ than SEE".

AUTHORS' RESPONSE: This sentence has been revised as suggested.

Line 98: change: evolved to a host-specific pathogen.to evolved to become a host-specific pathogen.

AUTHORS' RESPONSE: This sentence has been revised as suggested.

Line 101

This sentence is somewhat redundant. Change:

Comparisons of the accessory genome of the 50 SEE and 50 SEZ isolates were performed using the Spine, AGEnt, and ClustAGE pipeline (Fig. 1) to generate the AGEs identified among these isolates (Fig. 1).

To

Comparisons of the accessory genome of the 50 SEE and 50 SEZ isolates were performed using the Spine, AGEnt, and ClustAGE pipeline.

AUTHORS' RESPONSE: This sentence has been revised as suggested.

Line 109: What is SEQ_1102, why is it relevant in the context of this paragraph?

AUTHORS' RESPONSE: The manuscript has been revised to indicate that SEQ_1102 is a site-specific recombinase. It is reported here because it was identified as part of the AGEs.

Line 111: This wording is confusing. Please tighten.

Interestingly, all of the CDS that form each of the described ICE or prophage from SEE 4047 were not found in all our 50 SEE isolates

change to: "The CDSs associated with the ICE and prophages from SEE 4047 were not common to all 50 SEE isolates."

AUTHORS' RESPONSE: This sentence has been revised as suggested.

Line 114: you introduce the equibactin locus without informing the reader of what this is or why it's relevant. Some context here would be appreciated. I don't know what SEQ_1233 - SEQ_1246 are.

AUTHORS' RESPONSE: The manuscript was revised to explain that the equibactin locus is an ICE that is involved in iron acquisition, and to indicate the locus is encoded for by these 14 coding sequences, SEQ_1233 to SEQ_1246.

Line 118: change: and 23 CDS were characterized
to:
which, characterized 23 CDS.

AUTHORS' RESPONSE: This sentence has been revised as suggested.

Line 126: Tighten this sentence.

Change to: "Twelve of the fifteen SEZ-specific AGEs were localized to the cytoplasm (n = 5) or the cytoplasmic membrane (n = 7) in the bacterium, and single hypothetical protein was predicted to have an extracellular function."

AUTHORS' RESPONSE: This sentence has been revised as suggested.

Line 131, these sentences could be combined to be more succinct and less redundant.

The functions of the 15 CDS were evaluated using ClueGO, and a function was identified for only 3 CDS (Fig. 3, Table S2). Unsurprisingly, galactose metabolism was the only GO term described from the 3 CDS (lacE, lacF, and lacG).

Change to

"ClueGo evaluated the function of the 15 CDS, which characterized SZO_15220 - SZO_15250 and SZO_01750 as associated with galactose metabolism."

AUTHORS' RESPONSE: This sentence has been revised to be more succinct and less redundant.

We thank the reviewers for their helpful suggestions which have helped us to improve our report. We thank the reviewers and you for considering our revised manuscript. Please let me know if you have questions or concerns.

Sincerely,

Noah D. Cohen, VMD, MPH, PhD, Dipl. ACVIM (Large Animal)
Professor

December 5, 2021

Dr. Noah D Cohen
Texas A&M University
Large Animal Medicine & Surgery
College of Veterinary Medicine
Texas A&M University
College Station, TX 77843-4475

Re: Spectrum00764-21R1 (Differences in the accessory genomes and methylomes of strains of *Streptococcus equi* subsp. *equi* and of *Streptococcus equi* subsp. *zooepidemicus* obtained from the respiratory tract of horses from Texas)

Dear Dr. Noah D Cohen:

Link Not Available

Sincerely,

Cheryl Andam

Journals Department
Reviewer comments:

Reviewer #3 (Comments for the Author):

The clarity of the manuscript has been much improved. The authors addressed each review comment thoughtfully. I appreciate the additional analyses and discussion.

My only two comments:

- 1) Please revise sentences that say gene duplications lead to a loss of diversity. I'm not sure how this could be - duplications can lead to new functions/sub-functions or pseudogenes but either way, this would be an increase in diversity.
- 2) Please review for typos and syntax errors.

For example:

Discussion
line 372
"24 SEZ isolates but "
delete the "but"

line 385
change
"CDS methylated in SEE"
to
"CDSs methylated in SEE"

Line 385
change "indicates" to "suggests"

Staff Comments:

Preparing Revision Guidelines

Please return the manuscript within 60 days; if you cannot complete the modification within this time period, please contact me. If you do not wish to modify the manuscript and prefer to submit it to another journal, please notify me of your decision immediately so that the manuscript may be formally withdrawn from consideration by Microbiology Spectrum.

December 8, 2021

Professor Cheryl Andam
Editor, *Microbiology Spectrum*

Re: Spectrum00764-21 (Differences in the accessory genomes and methylomes of strains of *Streptococcus equi* subsp. *equi* and of *Streptococcus equi* subsp. *zooepidemicus* obtained from the respiratory tract of horses from Texas)

Dear Prof. Andam:

Thank you for your electronic message dated December 5, 2021 pertaining to the above-referenced manuscript. We thank the reviewers for their careful consideration of our report. We have revised the manuscript to address the comments from Reviewer 3. In this letter we provide a point-by-point response to each comment from the reviewer. Thank you for considering our responses and our revised report.

Comments from Reviewer 1

1) Please revise sentences that say gene duplications lead to a loss of diversity. I'm not sure how this could be - duplications can lead to new functions/sub-functions or pseudogenes but either way, this would be an increase in diversity.

AUTHORS' RESPONSE: The reviewer is correct, and we apologize for this error. We did not intend to indicate that gene duplications lead to reduced genetic diversity, but rather that gene duplications in SEE contribute to genomic differences between SEE and SEZ. Many of these duplications are the result of acquiring mobile genetic elements, and these acquired elements might contribute to the host specificity of SEE. We have revised the referent sentences in the manuscript accordingly.

2) Please review for typos and syntax errors.

AUTHORS' RESPONSE: I have reviewed the manuscript and attempted to correct all typographical and syntax errors. Respectfully, we defined the term CDS as plural so we disagree with the suggestion to use CDSs.

Thank you again for your assistance with the review of our manuscript. Please let me know if you have additional questions or concerns.

Sincerely,

Noah D. Cohen, VMD, MPH, PhD, DipI. ACVIM (Large Animal)
Professor

December 10, 2021

Dr. Noah D Cohen
Texas A&M University
Large Animal Medicine & Surgery
College of Veterinary Medicine
Texas A&M University
College Station, TX 77843-4475

Re: Spectrum00764-21R2 (Differences in the accessory genomes and methylomes of strains of *Streptococcus equi* subsp. *equi* and of *Streptococcus equi* subsp. *zooepidemicus* obtained from the respiratory tract of horses from Texas)

Dear Dr. Noah D Cohen:

Your manuscript has been accepted, and I am forwarding it to the ASM Journals Department for publication. You will be notified when your proofs are ready to be viewed.

Sincerely,

Cheryl Andam
Editor, Microbiology Spectrum

Journals Department
Supplemental Material: Accept
Supplemental Tables: Accept